# MondoA regulates gene expression in cholesterol biosynthesis-associated pathways required for zebrafish epiboly

Meltem Weger[1,2,3†]*, Benjamin D Weger[1,3,4†]*, Andrea Schink[1], Masanari Takamiya[1], Johannes Stegmaier[5‡], Cédric Gobet[4§], Alice Parisi[4], Andrei Yu Kobitski[1,6], Jonas Mertes[6#], Nils Krone[2¶], Uwe Strähle[1], Gerd Ulrich Nienhaus[1,6,7,8], Ralf Mikut[5], Frédéric Gachon[3,4], Philipp Gut[4], Thomas Dickmeis[1]*

[1]Institute of Biological and Chemical Systems – Biological Information Processing, Karlsruhe Institute of Technology, Eggenstein-Leopoldshafen, Germany; [2]Institute of Metabolism and Systems Research, College of Medical and Dental Sciences, University of Birmingham, Birmingham, United Kingdom; [3]Institute for Molecular Bioscience, The University of Queensland, Brisbane, Australia; [4]Nestlé Institute of Health Sciences SA, EPFL Innovation Park, Lausanne, Switzerland; [5]Institute for Automation and Applied Informatics, Karlsruhe Institute of Technology, Eggenstein-Leopoldshafen, Germany; [6]Institute of Applied Physics, Karlsruhe Institute of Technology, Karlsruhe, Germany; [7]Department of Physics, University of Illinois at Urbana-Champaign, Urbana, United States; [8]Institute of Nanotechnology, Karlsruhe Institute of Technology, Eggenstein-Leopoldshafen, Germany

*For correspondence:
m.weger@uq.edu.au (MW);
b.weger@uq.edu.au (BDW);
thomas.dickmeis@kit.edu (TD)

†These authors contributed equally to this work

Present address: ‡Institute of Imaging and Computer Vision, RWTH Aachen University, Aachen, Germany; §Institute of Bioengineering, School of Life Sciences, Ecole Polytechnique Fédérale de Lausanne, Lausanne, Switzerland; #Technobis tft-fos, Alkmaar, Netherlands; ¶Department of Oncology & Metabolism, University of Sheffield, Sheffield, United Kingdom

**Abstract** The glucose-sensing Mondo pathway regulates expression of metabolic genes in mammals. Here, we characterized its function in the zebrafish and revealed an unexpected role of this pathway in vertebrate embryonic development. We showed that knockdown of *mondoa* impaired the early morphogenetic movement of epiboly in zebrafish embryos and caused microtubule defects. Expression of genes in the terpenoid backbone and sterol biosynthesis pathways upstream of pregnenolone synthesis was coordinately downregulated in these embryos, including the most downregulated gene *nsdhl*. Loss of Nsdhl function likewise impaired epiboly, similar to MondoA loss of function. Both epiboly and microtubule defects were partially restored by pregnenolone treatment. Maternal-zygotic mutants of *mondoa* showed perturbed epiboly with low penetrance and compensatory changes in the expression of terpenoid/sterol/steroid metabolism genes. Collectively, our results show a novel role for MondoA in the regulation of early vertebrate development, connecting glucose, cholesterol and steroid hormone metabolism with early embryonic cell movements.

## Introduction

The glucose-sensing Mondo pathway is a key regulator of energy metabolism (*Abdul-Wahed et al., 2017*; *Richards et al., 2017*). It consists of three basic helix-loop-helix/leucine zipper (bHLH/Zip) family transcription factors (TFs): the two Mondo family factors 'Mlx interacting protein' (Mlxip; MondoA) and 'Mlx interacting protein-like' (Mlxipl; or ChREBP for 'carbohydrate response element binding protein') and their binding partner 'Max-like protein X' (Mlx) (*Havula and Hietakangas, 2018*; *Richards et al., 2017*). After their activation by glucose, likely via glucose-6-phosphate (G6P) (*Dentin et al., 2012*; *Li et al., 2010b*; *Stoltzman et al., 2008*), the Mondo family factors regulate

**eLife digest** In most animals, a protein called MondoA closely monitors the amount of glucose in the body, as this type of sugar is the fuel required for many life processes. Glucose levels also act as a proxy for the availability of other important nutrients. Once MondoA has detected glucose molecules, it turns genetic programmes on and off depending on the needs of the cell. So far, these mechanisms have mainly been studied in adult cells.

However, recent studies have shown that proteins that monitor nutrient availability, and their associated pathways, can control early development. MondoA had not been studied in this context before, so Weger et al. decided to investigate its role in embryonic development. The experiments used embryos from zebrafish, a small freshwater fish whose early development is easily monitored and manipulated in the laboratory.

Inhibiting production of the MondoA protein in zebrafish embryos prevented them from maturing any further, stopping their development at an early key stage. This block was caused by defects in microtubules, the tubular molecules that act like a microscopic skeleton to provide structural support for cells and guide transport of cell components.

In addition, the pathway involved in the production of cholesterol and cholesterol-based hormones was far less active in embryos lacking MondoA. Treating MondoA-deficient embryos with one of these hormones corrected the microtubule defects and let the embryos progress to more advanced stages of development.

These results reveal that, during development, the glucose sensor MondoA also controls pathways involved in the creation of cholesterol and associated hormones. These new insights into the metabolic regulation of development could help to understand certain human conditions; for example, certain patients with defective cholesterol pathway genes also show developmental perturbations. In addition, the work highlights a biological link between cholesterol production and cellular responses to glucose, which Weger et al. hope could one day help to identify new cholesterol-lowering drugs.

gene expression as MondoA:Mlx or ChREBP:Mlx heterodimers by binding E-box type enhancer elements ('carbohydrate response elements', 'ChoREs') in the regulatory regions of their target genes (*Billin et al., 2000*; *Jeong et al., 2011*; *Koo and Towle, 2000*; *Ma et al., 2006*; *Poungvarin et al., 2015*; *Yamashita et al., 2001*).

While recent studies have highlighted a crucial role for metabolism and metabolic signaling in embryonic development (*Miyazawa and Aulehla, 2018*), research on Mondo pathway function has concentrated on adult animals (*Hunt et al., 2015*; *Iizuka et al., 2004*; *Imamura et al., 2014*; *Song et al., 2019*). As it is currently unknown if and how MondoA, ChREBP or Mlx contribute to metabolic regulation of embryogenesis, we investigated a potential developmental role of the Mondo pathway in zebrafish, a model for early vertebrate development (*Nüsslein-Volhard, 2002*). After fertilization, a cap of cells (the blastoderm) forms on top of the yolk cell, which it subsequently engulfs in a morphogenetic movement called epiboly (*Bruce, 2016*). Mechanistically, epiboly involves processes both in the enveloping layer (EVL) as well as the deep cell layer (DEL) of the blastoderm and in a teleost specific structure of the yolk cell, the yolk syncytial layer (YSL). By the end of epiboly, germ layers and body axes have been formed. Most developmental pathways as well as many facets of metabolism and its regulation are highly conserved between mammals and zebrafish (*Gut et al., 2017*; *Schier and Talbot, 2005*; *Solnica-Krezel and Sepich, 2012*). However, studies addressing Mondo pathway function in the zebrafish are still lacking.

Herein, we uncover a role of the Mondo pathway in vertebrate development. Mondo pathway gene expression and function was characterized in zebrafish embryos and cultured cells. Morpholino oligonucleotide (MO) mediated loss-of-function of MondoA severely impaired epiboly movements. Subsequent transcriptome analysis revealed genes deregulated upon loss of MondoA function and highlighted an enzyme involved in cholesterol biosynthesis, Nsdhl (NAD(P) dependent steroid dehydrogenase-like), as a potential key mediator of MondoA function in epiboly. Functional analysis of Nsdhl suggests that Nsdhl-mediated cholesterol synthesis downstream of MondoA is required for the synthesis of sufficient levels of the steroid hormone pregnenolone, which stabilizes YSL

microtubules necessary for epiboly. Maternal-zygotic (MZ) mutant embryos homozygous for a small deletion allele of *mondoa* leading to a premature stop codon showed a severe aberrant epiboly phenotype, albeit with low penetrance that our transcriptome analysis indicates to result from compensatory changes in the expression of cholesterol and steroid biosynthesis genes.

## Results

### Glucose signaling by the Mondo pathway is conserved in zebrafish

For each of the Mondo pathway members MondoA, ChREBP and Mlx one single orthologue is present in the zebrafish genome (GRCz11/danRer11). We cloned the full cDNAs for zebrafish MondoA and ChREBP (GenBank Accession KF713493 [*mondoa*], KF713494 [*chrebp*]) based on the (partially) predicted sequences. Phylogenetic analysis showed that zebrafish *mondoa*, *chrebp* and *mlx* cluster with their mammalian and chicken homologs (*Figure 1A*) and revealed a high level of protein sequence conservation between zebrafish and human orthologs (*Figure 1B*), especially of the glucose-sensing module (GSM) specific to the Mondo family and of the DNA binding bHLH/Zip domains. This finding suggests that the functions of these proteins are conserved.

To examine whether the zebrafish Mondo pathway factors function similarly in the regulation of gene transcription as their mammalian orthologs, we studied the pathway in zebrafish PAC2 cells (*Lin et al., 1994*). All three Mondo pathway factors are expressed in these cells (*Figure 1—figure supplement 1A*). To monitor Mondo signaling, we generated a luciferase reporter gene construct driven by two ChoREs fitting the mammalian consensus (*Ma et al., 2006*) and a TATA box minimal promoter (*Figure 1C*). This construct was active in HepG2 cells, a mammalian cell culture model commonly used for Mondo pathway studies (*Kim et al., 1996*; *Yu and Luo, 2009*; *Figure 1—figure supplement 1B*). In PAC2 cells, bioluminescence equally increased in a dose-dependent manner upon glucose treatment (*Figure 1C*). No significant changes in bioluminescence levels were detected upon glucose treatment in cells expressing a constitutively active luciferase reporter (pGL3-Control; *Figure 1D*), excluding a general unspecific increase in transcriptional activity by glucose treatment.

We next tested whether overexpression of Mondo pathway members enhances glucose induced pathway activity. Transient overexpression of MondoA and Mlx activated transcription from the reporter also under low glucose conditions (*Figure 1E*), as shown by the increased bioluminescence compared with transfection of the 2xChoRE reporter alone ($p \leq 0.001$). Overexpression of the MondoA and Mlx factors together led to an even more pronounced effect on bioluminescence ($p \leq 0.01$), revealing synergistic effects. High glucose levels caused significant reporter gene induction in control cells ($p \leq 0.01$; *Figure 1E*). Importantly, strong glucose induction of the reporter was also shown by cells overexpressing either Mlx ($p \leq 0.01$) or MondoA ($p \leq 0.001$) alone as well as both factors together ($p \leq 0.001$; *Figure 1E*). An unrelated control reporter construct (pGRE-Luc, *Weger et al., 2012*) was not responsive to any of the treatments (*Figure 1—figure supplement 1C*). Together, the data indicate that limited availability of endogenous pathway components restrains pathway activity, and that both basal activity and the response to higher glucose levels are potentiated when more sensor proteins are available.

In addition, we explored the effect of morpholino oligonucleotide (MO) mediated loss-of-function of *mondoa* to confirm that Mondo pathway function is required for glucose induction of reporter expression. Cells transfected with a control 5 bp mismatch MO (mondoa-mis) showed a 2.0-fold induction of bioluminescence from the 2xChoRE reporter plasmid by high glucose levels, while in cells transfected with a MO directed against the translation start site (mondoa-mo) this glucose response was abolished (*Figure 1F*). Taken together, our data demonstrate that ChoRE mediated glucose induction of gene expression is regulated by the Mondo pathway also in zebrafish.

### The zebrafish Mondo pathway is present in early embryos

After confirming the similarity of zebrafish Mondo pathway function to mammals in cultured cells, we studied its role during development. We reanalyzed a previously published zebrafish developmental transcriptome dataset (*White et al., 2017*, ENA accession number ERP014517) and detected maternal transcripts of *mondoa*, *chrebp* and *mlx* as well as expression of the three genes throughout development, following distinct temporal patterns (*Figure 1G–I*). Whole mount *in situ* hybridization

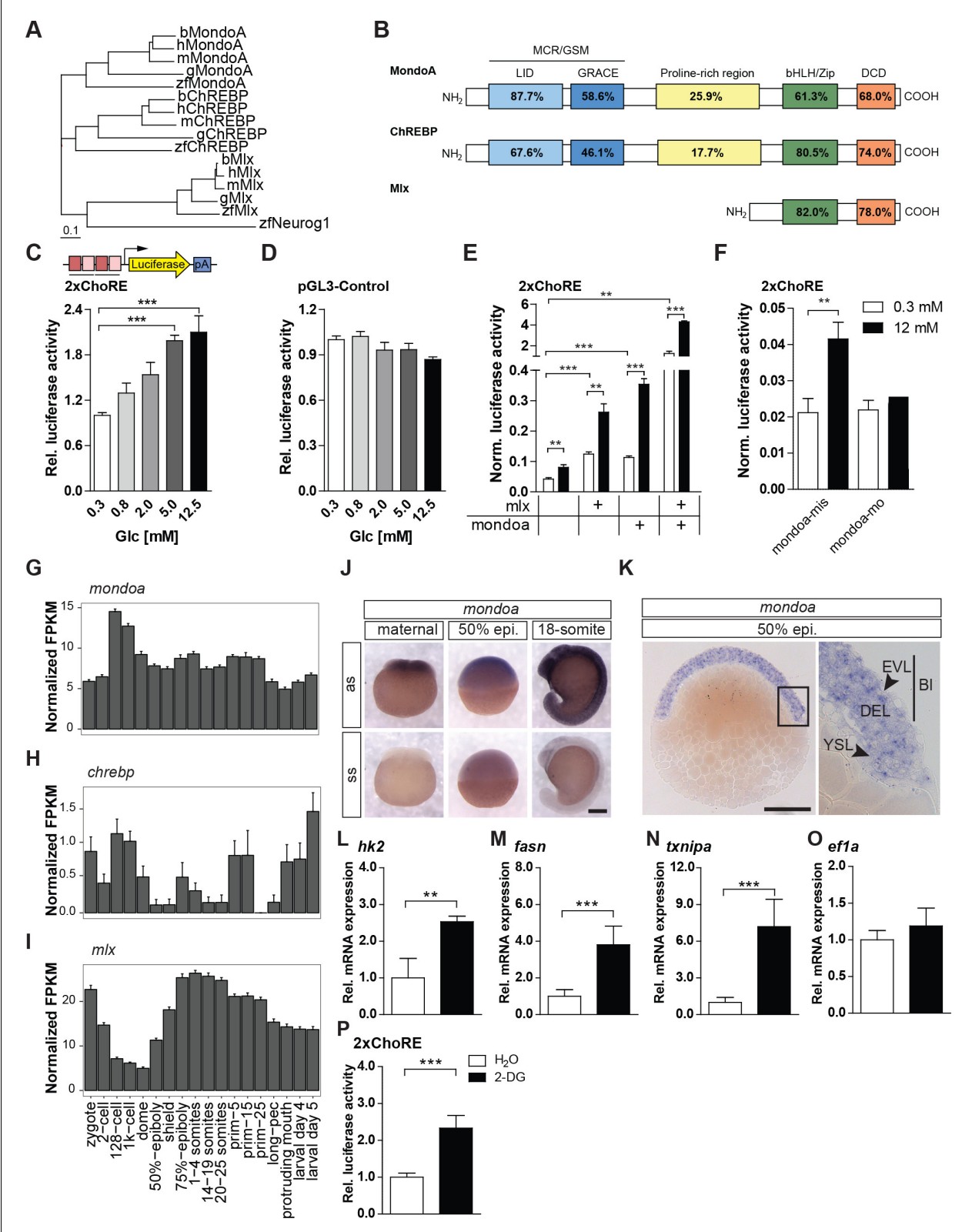

**Figure 1.** The glucose-sensing Mondo pathway in zebrafish. (**A**) Phylogenetic tree of ChREBP, MondoA and Mlx proteins. h, human, m, mouse, b, bovine, g, chicken, zf, zebrafish. Outgroup: zf Neurogenin 1 (Neurog 1). Scale bar: 0.1 estimated amino acid substitutions per site. (**B**) Amino acid identities in % between zebrafish and human domains of ChREBP, MondoA and Mlx: 'Mondo conserved regions/glucose-sensing module' (MCR/GSM); 'low-glucose inhibitory domain' (LID; light blue); 'glucose-response activation conserved element' (GRACE; dark blue); 'basic-helix-loop-helix/leucine

*Figure 1 continued on next page*

Figure 1 continued

zipper' (bHLH/ZIP; green); 'dimerization and cytoplasmic localization domain' (DCD; red). (C, D) Bioluminescence levels after 24 hr of glucose treatment of PAC2 cells transiently transfected with the 2xChoRE reporter consisting of a luciferase reporter gene (yellow) regulated by a minimal promoter (TATA; arrow) and two carbohydrate response elements (ChoREs; each with the sequence 5'-CACGCG-N5-CTCGTG-3'; pA for polyadenylation site; n = 4, (C) or with the constitutively expressed pGL3-Control reporter construct (D, n = 4). (E, F) Bioluminescence levels in 2xChoRE reporter expressing PAC2 cells upon 24 hr of treatment with 0.3 mM (white bars) or 12 mM (black bars) glucose after overexpression of MondoA and/or Mlx (E, n = 4) or transfection with mondoa-mo or mondoa-mis (F, n = 8). Data were normalized to Renilla luciferase activity (Norm. bioluminescence). (G–I) mRNA expression profiles of *mondoa* (G), *chrebp* (H) and *mlx* (I) during zebrafish developmental stages from zygote to larval stage five extracted from a published dataset (**White et al., 2017**). n = 20; FKPM, Fragments Per Kilobase of transcript per Million mapped reads. (J) WISH of *mondoa* transcripts at zygote (maternal), 50% epiboly (50% epi.) and 18-somite stages. as, antisense probe, ss, sense probe. Scale bar: 0.2 mm. (K) Epon sections of 50% epi. embryos showed *mondoa* expression in the enveloping layer (EVL), the deep cell layer (DEL) of the blastoderm (Bl) and the yolk syncyctial layer (YSL). Scale bar: 0.2 mm, for higher magnification 50 μm. (L–O) Glucose induction of Mondo pathway target gene expression in early zebrafish embryos. Embryos were injected with the glucose analog 2-deoxy-D-glucose (2-DG; black bars) or with water (white bars) as a control. RNA was extracted at the sphere stage to perform RT-qPCR of genes known to be Mondo pathway targets in mammals: *hexokinase 2* (*hk2*, L), *fatty acid synthase* (*fasn*, M), *thioredoxin-interacting protein a* (*txnipa*, N), *eukaryotic translation elongation factor 1 alpha 1* (*ef1a*, O) (n = 9). (P) Embryos (n ≥ 72) injected with the 2xChoRE reporter showed increased bioluminescence at sphere stage when co-injected with 2-DG. Error bars represent SEM; *, p≤0.05; **, p≤0.01; ***, p≤0.001.

The online version of this article includes the following figure supplement(s) for figure 1:

**Figure supplement 1.** Additional data of Mondo pathway characterization experiments in zebrafish.

analysis (WISH) for *mondoa* revealed a ubiquitous expression pattern (**Figure 1J**). Similar results were observed for *chrebp* and *mlx* (**Figure 1—figure supplement 1D,E**). At the 50% epiboly stage, all three genes are expressed throughout the embryo (**Figure 1K**; **Figure 1—figure supplement 1F**).

Next, we tested whether glucose treatment increases the expression of mammalian Mondo pathway target gene homologs in early zebrafish embryos, namely *hexokinase 2* (*hk2*) (**Sans et al., 2006**), *fatty acid synthase* (*fasn*) (**Ma et al., 2006**) and *thioredoxin-interacting protein a* (*txnipa*) (**Stoltzman et al., 2008**). We injected the glucose analogue 2-deoxy-glucose (2-DG), which is metabolized to the G6P analogue 2-DG6P but not further (**Chi et al., 1987**; **Stoltzman et al., 2008**), thereby avoiding activation of other pathways relying on downstream metabolization of glucose. This treatment significantly increased expression of *hk2* (p<0.001), *fasn* (p<0.01) and *txnipa* (p<0.001; **Figure 1L–N**), but not of the control gene *eef1a1l1/ef1a* (*eukaryotic translation elongation factor 1 alpha 1* (**Figure 1O**), ruling out a nonspecific general increase in gene expression by 2-DG treatment. Furthermore, *txnipa* expression was not induced by 2-DG in embryos injected with mondoa-mo, showing that MondoA function is required for its induction (**Figure 1—figure supplement 1G**). To directly examine if 2-DG treatment regulates transcription by ChoRE enhancer elements in the early embryo, we injected the 2xChoRE luciferase reporter construct together with 2-DG. Embryos treated with 2-DG (n = 72) showed a 2.3-fold increase (p≤0.01) in bioluminescence over the control (n = 78; **Figure 1P**). Together, these results suggest that Mondo pathway mediated glucose signaling is present in embryos as early as at the sphere stage.

## Loss of *mondoa* function leads to severe epiboly defects

To examine Mondo pathway function during development, we injected embryos with MOs targeting *mondoa* and *chrebp* function. While injections of a MO directed against *chrebp* did not result in an aberrant phenotype (**Figure 2—figure supplement 1A,B**), mondoa-mo injected embryos showed a striking delay in epiboly movements when compared with uninjected or mondoa-mis injected embryos (**Figure 2A–D**; **Video 1**). This phenotype was dose-dependent, with the strongest delay occurring at the highest concentration (**Figure 2A**).

To test for efficiency and specificity of the MO-mediated knockdown, a number of controls were employed. GFP expression from a reporter construct carrying the target sequence was efficiently abolished by co-injection of mondoa-mo, but not mondoa-mis (**Figure 2—figure supplement 1C**). An involvement of P53-mediated apoptosis in the observed phenotypes was ruled out by co-injection of mondoa-mo and a MO directed against *p53* (p53-mo), leading to the same phenotype as observed upon injection with mondoa-mo alone (**Figure 2—figure supplement 1D**). Knockdown of the *mondoa* cofactor *mlx* with a MO directed against the translation start site of *mlx* (mlx-mo)

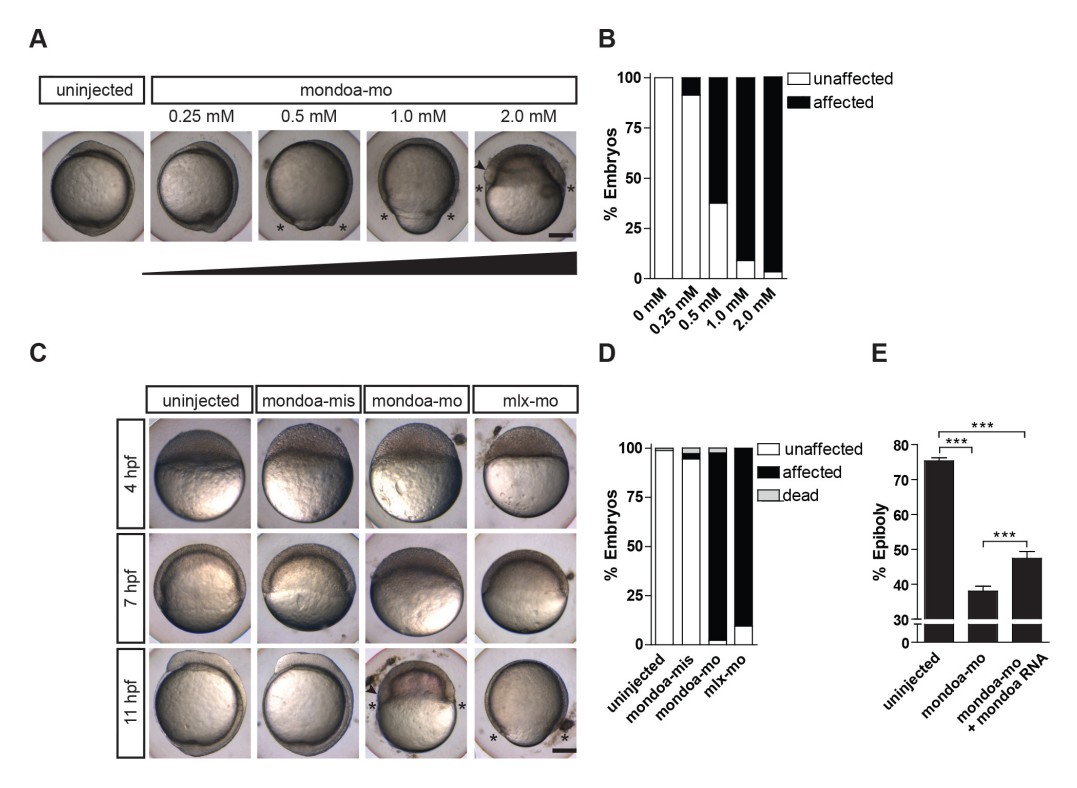

**Figure 2.** Knockdown of *mondoa* impacts zebrafish epiboly. (**A, B**) Dose-dependent (0.25–2.0 mM) arrest of epiboly movements of *mondoa* morphants when the uninjected control was around bud stage (~10 hpf). (**A**) Images of representative embryos. (**B**) Quantification of phenotypes scored when uninjected embryos had accomplished epiboly: 'unaffected', embryos reached bud/3 somite stage in parallel with controls; 'affected', developmental arrest at an earlier epiboly stage; 'dead', coagulated embryos. Percentages of 'affected' embryos were: uninjected (n = 0/26; 0%), 0.25 mM (n = 2/23; 8.7%), 0.5 mM (n = 15/24; 62.5%), 1 mM (n = 17/18; 94.4%), 2 mM (n = 39/41; 95.1%). (**C**) Developmental time course upon *mondoa* knockdown compared to uninjected embryos at the sphere stage (~4 hpf), the 60% epiboly (~7 hpf) and the 3-somite stage (~11 hpf). When uninjected control embryos were at the 3-somite stage, the epiboly delay/arrest phenotype was observed with mondoa-mo and with a MO targeting mlx (mlx-mo). No effect on epiboly was observed in embryos injected with mondoa-mis MO. (**D**) Quantification of the phenotypes in (**C**), scored when uninjected embryos were at the bud stage. Percentages of 'affected' embryos : uninjected, 0% (n = 0/403); mondoa-mis, 2.8% (n = 3/109); mondoa-mo, 95.3% (n = 462/485); mlx-mo, 90.5% (n = 124/137). (**E**) Partial rescue of *mondoa* knockdown with *mondoa* mRNA. Percentage of epiboly progression was determined via *no tail* (*ntl*) WISH. Uninjected control embryos (n = 30) were fixed at the 75% epiboly stage along with the morphants. Embryos co-injected with mondoa-mo and *mondoa* mRNA (250 ng/μl) were partially rescued (n = 20) compared with mondoa-mo alone injected embryos (n = 25). Scale bar: 0.2 mm. Asterisks label the blastoderm margin, arrowhead indicates germ ring/shield-like thickening. Error bars = SEM; *, p≤0.05; **, p≤0.01; ***, p≤0.001. hpf, hours post fertilization.

The online version of this article includes the following figure supplement(s) for figure 2:

**Figure supplement 1.** MO-mediated knockdown of *chrebp* and additional *mondoa* MO control experiments.

showed a dose-dependent impairment of epiboly similar to the one observed with mondoa-mo, albeit somewhat less severe (*Figure 2C,D*). These results are consistent with the model that MondoA forms heterodimers with Mlx to exert its function also in epiboly. Finally, to obtain additional evidence that the observed effects are indeed dependent on MondoA function, we attempted to rescue epiboly movements in mondoa-mo injected embryos by co-injecting a *mondoa* mRNA in which the sequence targeted by mondoa-mo was mutated. Epiboly progression was monitored with WISH against the blastoderm margin marker *no-tail* (*ntl*). Embryos injected with only mondoa-mo reached 38% epiboly, whereas embryos co-injected with *mondoa* mRNA arrived at 47% epiboly (*Figure 2E*). These values are consistent with those reported in rescue experiments addressing other genes implicated in epiboly (*Hsu et al., 2006*; *Schwend et al., 2011*) and might be explained by decreased mRNA stability and/or decreased translation efficiency of the rescue construct *in vivo*. In conclusion, these observations indicate that mondoa-mo specifically targets *mondoa* gene function, and reveal that loss of MondoA function severely impacts on epiboly during zebrafish development.

## Loss of *mondoa* function specifically affects YSL epiboly

Next, we aimed at clarifying which epiboly-related processes are perturbed by loss of *mondoa* function. Since the Mondo pathway has been implicated in the regulation of energy metabolism in adult mammalian tissues, we tested if the aberrant epiboly phenotype of *mondoa* morphants is a consequence of generally impaired energy metabolism. Our results showed that energy charge was unaltered in the morphants (*Figure 3—figure supplement 1A*). Alternatively, abnormal cell proliferation or defects in migratory behavior might limit the spreading capacity of the blastoderm. To count cell numbers in the embryos by automated quantification, we imaged transgenic embryos carrying a H2A transgene which labels cell nuclei (*Tg(h2afva:h2afva-GFP)*) (*Pauls et al., 2001*). We employed digital scanned laser light sheet microscopy (DSLM) (*Kobitski et al., 2015*), allowing the optical sectioning of the embryo with a high spatial and temporal resolution. As shown by this analysis, *mondoa* morphants had similar numbers of cells as control embryos (*Figure 3A*). Accordingly, the cell density was higher in *mondoa* morphants compared to mismatch controls, as movement to the vegetal pole was impaired while cells continued to divide (*Figure 3B,C*; *Videos 2–4*). These observations strongly argue against deficient cell proliferation. In addition, we observed germ ring-like thickenings (*Figure 2A,C*; arrowheads) as well as internalization of cells at the blastoderm margin in *mondoa* morphants (*Figure 3D*; *Figure 3—figure supplement 1B–D*; *Videos 5* and *6*). Furthermore, the average migration speed of deep cells was identical for morphants and controls, demonstrating normal individual cell migration capacity in morphants (*Figure 3—figure supplement 1E*).

EVL, DEL and YSL have all been implicated in mechanisms of epiboly (*Bruce, 2016*). Thus, we turned to examining how the three different embryonic cell layers were affected in *mondoa* morphants. Imaging of *Tg(h2afva:h2afva-GFP)* embryos allowed us to assign nuclei to the EVL or to the DEL and to distinguish the YSL nuclei. Our observations revealed that EVL cells continued to progress and could reach a position vegetally to the YSL nuclei in the morphants (*Figure 4—figure supplement 1A–F*, *Video 7*). Consistently, EVL integrity as revealed by rhodamine-phalloidin staining (*Lepage and Bruce, 2010*; *Lepage et al., 2014*) was intact in the morphants (*Figure 4—figure supplement 1G–H*). These findings indicate that EVL epiboly still occurs in the morphants.

However, deep cells of *mondoa* morphants lagged behind both EVL and YSL layers, consistent with models suggesting rather passive movement of these cells into spaces left open by the other two layers (*Bensch et al., 2013*; *Song et al., 2013*; *Figure 4A,B*; *Figure 4—figure supplement 1A–F*). Nevertheless, in *mondoa* morphants a few deep cells were detected more vegetally than the YSL nuclei. Also, YSL nuclei showed a much broader and more disorganized distribution than in control injected or wild-type embryos (*Figure 4—figure supplement 1A–F*). The abnormal shape of the YSL was already evident at a slightly earlier stage (compare *Figure 4A* with *Figure 4B*). YSL nuclei appeared to cluster and to break away from the blastoderm margin (*Figure 4B*, white arrow), in contrast to the more even distribution in a relatively narrow band around the margin in wild-type

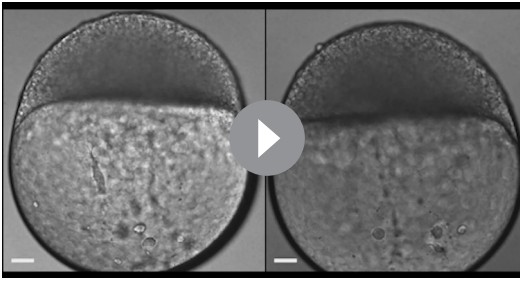

**Video 1.** Time-lapse video microscopy of a wild-type embryo and a *mondoa* morphant (brightfield). Lateral view, animal pole is on top. Wild-type embryo (left), *mondoa* morphant (right). Imaging was performed at an ambient temperature of 26°C. Time between frames 2.1 min. Scale bar: 65 μm.
https://elifesciences.org/articles/57068#video1

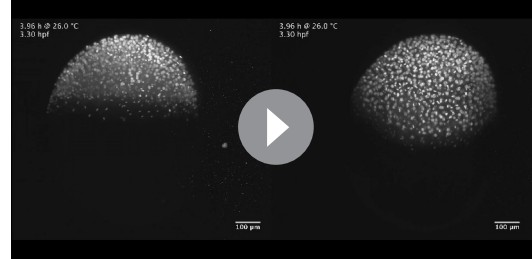

**Video 2.** DSLM time-lapse video microscopy of epiboly of a mondoa-mis and a mondoa-mo injected *Tg(h2afva:h2afva-GFP)* embryo. Lateral view, animal pole is on top. mondoa-mis (left), mondoa-mo (right). Imaging was performed at an ambient temperature of 26°C. Time between frames 1.0 min. Scale bar: 100 μm.
https://elifesciences.org/articles/57068#video2

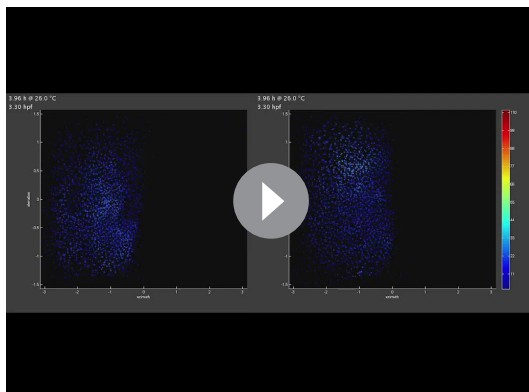 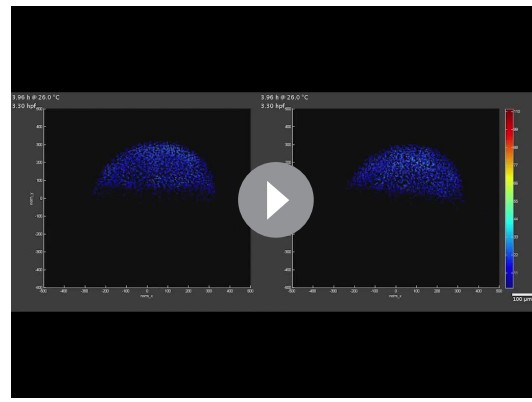

**Video 3.** Cell density plot of a mondoa-mis and a mondoa-mo injected *Tg(h2afva:h2afva-GFP)* embryo (2D projection of azimuth and elevation angles). 2D projection using spherical coordinates of the origin centered nuclei positions of a mondoa-mis (left) and a mondoa-mo (right) injected embryo. Azimuth and elevation angles are plotted on the x and y axis, respectively, and the cell density is indicated by the color code (transition from blue to red indicates increasing density). Animal pole is on the left.
https://elifesciences.org/articles/57068#video3

**Video 4.** Cell density plot of a mondoa-mis and a mondoa-mo injected *Tg(h2afva:h2afva-GFP)* embryo (projection of x and y Cartesian coordinates). Projection using Cartesian x and y coordinates of the origin centered nuclei positions of a mondoa-mis (left) and a mondoa-mo (right) injected embryo. The cell density is indicated by the color code (transition from blue to red indicates increasing density). Animal pole is on the top.
https://elifesciences.org/articles/57068#video4

embryos. These findings pointed to a defect in YSL structure or function as a potential reason for the impaired epiboly in *mondoa* morphants.

To test whether MondoA function in the YSL itself is important for epiboly, we injected mondoa-mo at the 1 k-cell stage into the yolk cell only (*Figure 4C*). At this late time of YSL injection, additional *mondoa* transcript and protein have been produced after mid-blastula transition (MBT), which may limit knockdown efficiency compared with injection into the zygote. Strikingly, *mondoa* knockdown limited to the YSL still led to epiboly delay and developmental arrest compared with uninjected or control injected embryos (*Figure 4D*). Thus, MondoA function in the YSL indeed is important for correct epiboly movements. Impaired YSL structure or function may lead to impaired vesicle trafficking at the YSL-EVL border (*Lepage and Bruce, 2010*; *Solnica-Krezel et al., 1995*; but see also *Lepage and Bruce, 2014*) or to defective patterning of the overlying blastoderm (*Carvalho and Heisenberg, 2010*), both of which can perturb epiboly. However, we did not find any evidence for major perturbations of marginal endocytosis or blastoderm patterning in the morphants (*Figure 4—figure supplement 1J–Q*).

### Transcriptome analysis points to the cholesterol synthesis enzyme *nsdhl* as a main target gene of MondoA in the early embryo

To obtain cues how MondoA function might impact on epiboly we carried out an unbiased examination of differential gene expression upon *mondoa* knockdown by applying RNA-seq. Embryos were injected either with mondoa-mo or mondoa-mis and left to develop until the control embryos reached sphere stage, just before the morphological phenotype becomes apparent (*Figure 5—figure supplement 1A–C*; *Supplementary file 1*). We validated a random subset of the genes expressed differentially between the two conditions by RT-qPCR, confirming the RNA-seq data (*Figure 5—figure supplement 1D,E*). Interestingly, the gene most down-regulated by *mondoa*-knockdown was *nsdhl* (*NAD(P) dependent steroid dehydrogenase-like*; *Figure 5A*). Nsdhl is part of the cholesterol biogenesis pathway, catalyzing steps in the conversion of the cholesterol precursor lanosterol to zymosterol (*Figure 5B*). Consistently, a global metabolic pathway gene set enrichment analysis revealed that both the 'sterol biosynthesis' pathway, of which Nsdhl is part, and the upstream 'terpenoid backbone biosynthesis' pathway are coordinately downregulated in the morphants (*Supplementary file 2*, *Figure 5B*, *Figure 7—figure supplement 1A,B*).

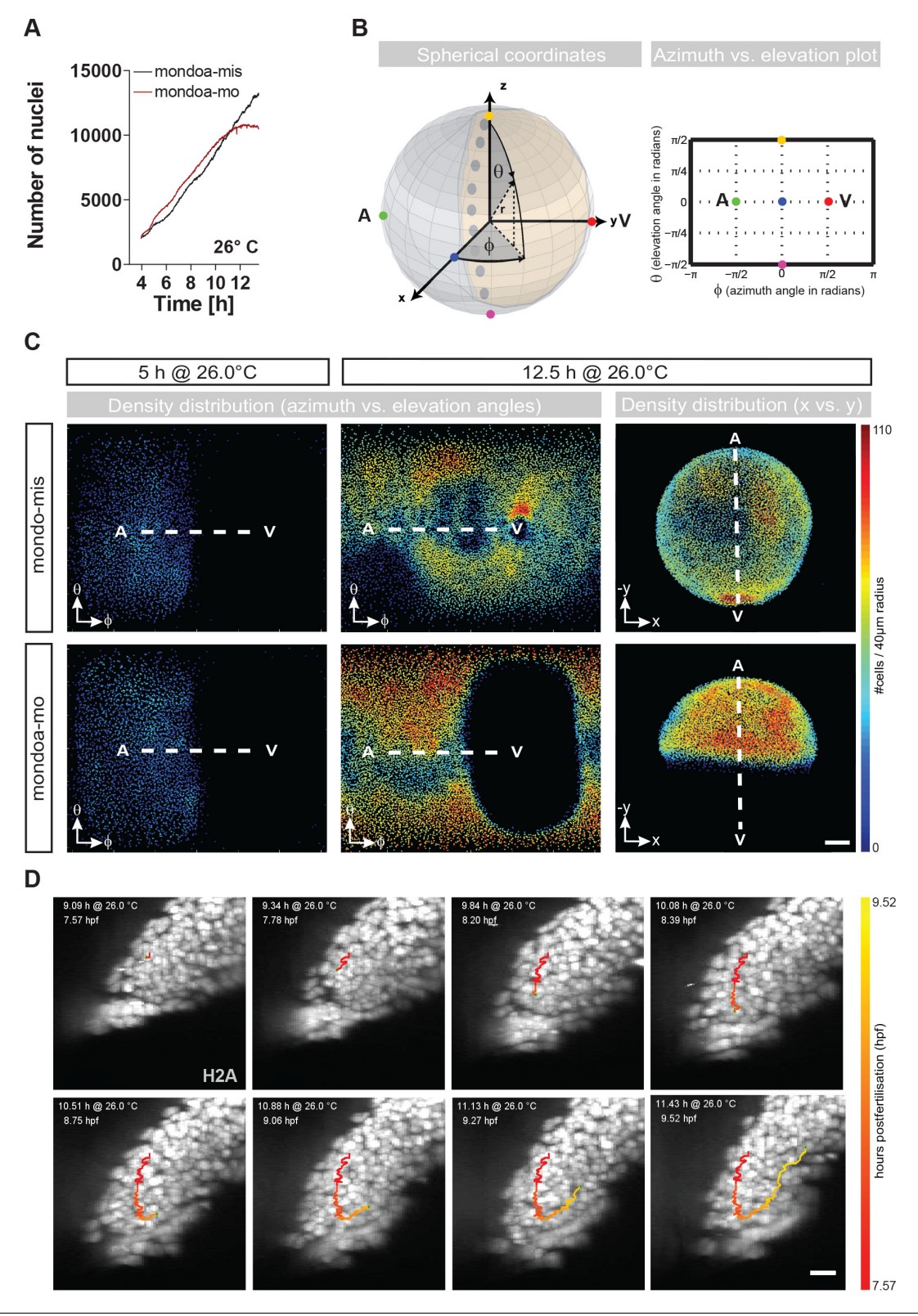

**Figure 3.** *mondoa* knockdown does not impair cell division or internalization movements. (**A**) Nuclei numbers over time in hours of mondoa-mis (black trace) and mondoa-mo (red trace) injected *Tg(h2afva:h2afva-GFP)* embryos. (**B**) Scheme illustrating the transformation of Cartesian coordinates of nuclei positions extracted from the DSLM image data into a two-dimensional plot of azimuth vs. elevation angles. For details, see Materials and methods section. (**C**) Cell density plots of mondoa-mis and mondoa-mo injected embryos after 5 and 12.5 hr of imaging using an azimuth vs. elevation angle plot

*Figure 3 continued on next page*

*Figure 3 continued*

to unwrap the embryos. For comparison, a projection of x and y Cartesian coordinates is also shown for the 12.5 hr data. Cell density is color-coded (number of cells/40 μm radius). (D) DSLM time lapse video stills from *Video 5* of a morphant embryo. The trace of a single cell determined by automated cell nucleus tracking is highlighted (red to yellow). Scale bar, 20 μm.

The online version of this article includes the following figure supplement(s) for figure 3:

**Figure supplement 1.** Energy charge in sphere stage embryos as well as internalization and movement speed of deep cells are not affected in mondoa morphants.

## *nsdhl* expression is regulated by the Mondo pathway in a glucose-dependent manner and loss of its function perturbs epiboly

Cholesterol has an important role as substrate for the synthesis of pregnenolone, the first steroid hormone within the steroid hormone biosynthesis pathway. Pregnenolone and other steroids have been implicated in the regulation of zebrafish epiboly (*Eckerle et al., 2018*; *Hsu et al., 2006*; *Schwend et al., 2011*; *Weng et al., 2013*). Therefore, we hypothesized that *mondoa* knockdown might affect epiboly by interfering with Nsdhl-dependent cholesterol and steroid hormone biogenesis. To test this hypothesis, we first validated regulation of *nsdhl* expression by MondoA mediated glucose sensing. We observed a significant increase of basal *nsdhl* transcript levels upon 2-DG injection ($p \leq 0.05$) and upon overexpression of both MondoA and Mlx ($p \leq 0.001$) in early embryos, which was augmented further when both treatments were combined ($p \leq 0.001$; *Figure 5C*). No changes in expression were observed for *ef1a* under these conditions (*Figure 5D*), confirming the target gene specific effects of the treatments. Together, these results strongly suggest that the Mondo pathway mediates glucose regulation of *nsdhl* expression, thereby linking glucose sensing with cholesterol synthesis.

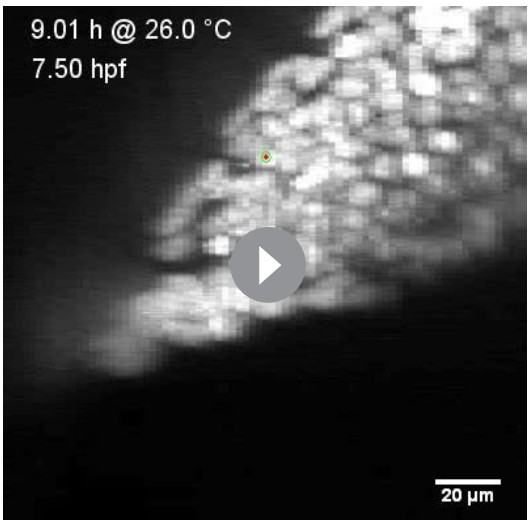

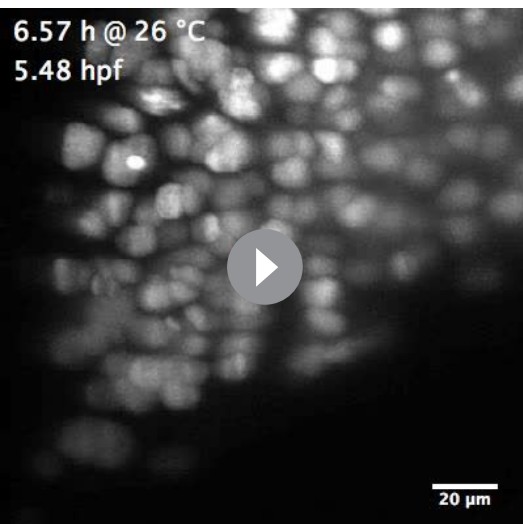

**Video 5.** DSLM time-lapse video microscopy with cell tracking of the blastoderm edge of a *mondoa* morphant *Tg(h2afva:h2afva-GFP)* embryo. Lateral view, animal pole is on top. Imaging was performed at an ambient temperature of 26°C. Time between frames 1.0 min. The trace of an individual cell determined by automated cell nucleus tracking is highlighted (red to yellow), revealing its internalization and migration towards the animal pole despite epiboly arrest. Scale bar: 20 μm.

https://elifesciences.org/articles/57068#video5

**Video 6.** DSLM time-lapse video microscopy with cell tracking of the blastoderm edge of a *mondoa*-mis–injected control *Tg(h2afva:h2afva-GFP)* embryo. Lateral view, animal pole is on top. Imaging was performed at an ambient temperature of 26°C. Time between frames 1.0 min. The trace of an individual cell determined by automated cell nucleus tracking is highlighted (red to yellow), revealing its internalization and migration towards the animal pole despite epiboly arrest. Scale bar: 20 μm.

https://elifesciences.org/articles/57068#video6

We next examined whether *nsdhl* knockdown can affect epiboly using a MO targeted against a splice site of the *nsdhl* gene (nsdhl-mo). We observed a highly similar phenotype to the *mondoa* knockdown phenotype, with nsdhl-mo injected embryos arrested at 50% epiboly when control embryos had completed epiboly (*Figure 5E,F*). As *mondoa*, *nsdhl* is ubiquitously expressed in the embryo, including the YSL (*Figure 5—figure supplement 1F,G*), and injection of nsdhl-mo into the yolk cell at the 1 k-cell stage resulted in an epiboly delay and developmental arrest as observed for YSL specific *mondoa* knockdown (*Figure 5G*). Thus, the function of Nsdhl in the biosynthesis pathway to cholesterol in the YSL appears to be essential for epiboly movements.

## Pregnenolone partially rescues epiboly in *mondoa* morphants and normalizes microtubule patterns

Altered cholesterol biosynthesis in the YSL by loss of MondoA/Nsdhl function might affect steroid hormone-dependent epiboly mechanisms. Therefore, we next tested whether pregnenolone treatment can rescue *mondoa* morphants. Treatment with pregnenolone indeed partially rescued epiboly in mondoa-mo injected embryos (*Figure 5H*) at a level comparable to the rescue with *mondoa* mRNA. Accordingly, the aberrant epiboly phenotype of *nsdhl* morphants was also partially rescued by pregnenolone treatment (*Figure 5I*), confirming its role downstream of the MondoA/Nsdhl pathway.

Pregnenolone was shown to regulate epiboly by mediating microtubule stability (*Hsu et al., 2006*; *Schwend et al., 2011*; *Weng et al., 2013*). To examine whether yolk cell microtubule integrity is affected in *mondoa* morphants, we stained embryos against α-tubulin. In uninjected embryos (n = 9/9), the YSL forms a regular band below the blastoderm, with yolk syncytial layer nuclei (YSN) surrounded by microtubule organizing centers (MTOCs) (*Solnica-Krezel and Driever, 1994*; *Strähle and Jesuthasan, 1993*; *Figure 6A*). Below this area, dense arrays of microtubules spread from the MTOCs along the animal-vegetal axis into the yolk cytoplasmic layer (YCL), forming root-like bundles. Compared with the control embryos (*Figure 6C,D*), mondoa-mo injected embryos showed a rather unstructured YSL with a less well defined border (*Figure 6E,F*). YSN were less regularly arranged, and their associated MTOCs formed stellar structures (n = 14/16; *Figure 6B*), apparently reflecting shortened microtubules that did not form the arrays of root-like bundles along the animal-vegetal axis seen in uninjected controls. Importantly, the YSL of morphants treated with pregnenolone appeared to be better organized compared to the untreated morphants because the YSN and their corresponding MTOCs were arranged as in the uninjected control (n = 6/9; *Figure 6B,G,H*). Furthermore, in pregnenolone treated embryos, the microtubule arrays were longer and formed array-like structures along the animal-vegetal axis of the embryos similar to the control. Taken together, these results strongly suggest that MondoA functions in epiboly via the stabilization of yolk microtubules by pregnenolone formed downstream of cholesterol biosynthesis.

## Maternal-zygotic mutants of *mondoa* show partially penetrant aberrant epiboly phenotypes

Finally, we wanted to validate the knockdown results with a genetic loss-of-function model of *mondoa* using CRISPR-Cas9. We identified a mutant allele transmitted through the germline that carried a 5 bp deletion in exon 3 (*Figure 7A*), leading to a frameshift in the coding sequence and a predicted premature stop eliminating both the DNA binding domain and most of the glucose-sensing

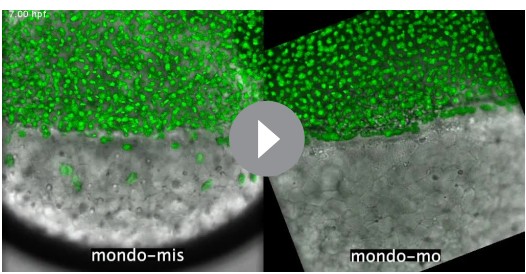

**Video 7.** EVL epiboly is not affected in mondoa morphants. *Tg(h2afva:h2afva-GFP)* embryos were injected at the one-cell stage with either mondoa-mis (left) or mondoa-mo (right). The epiboly phenotype was examined by confocal microscopy from 7 to 15 hpf (at 28°C). Time frames were acquired every 13 min with a 2 μm z-step size to cover ~180 μm in depth. EVL nuclei of a *mondoa* morphant (right), identified by the confocal reflection channel (not shown), were indicated by white circles at 10.08 hpf, the time when a mondoa-mis-injected embryo (left) reached bud stage. Note the presence of EVL nuclei (white circles) located vegetal to the blastoderm margin, indicative of unaffected EVL epiboly in *mondoa* morphants.
https://elifesciences.org/articles/57068#video7

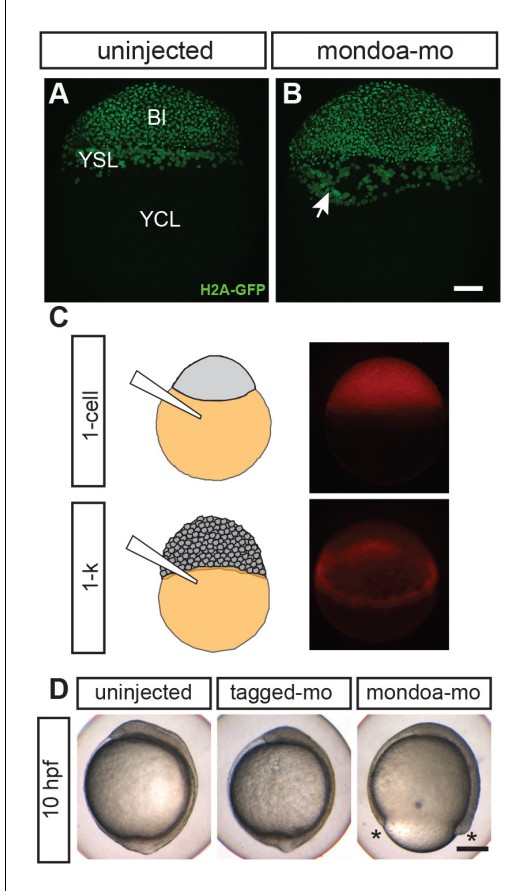

**Figure 4.** MondoA function in the YSL contributes to epiboly. (**A, B**) Confocal images of *Tg(h2afva:h2afva-GFP)* embryos injected with mondoa-mo (B, n = 5) and uninjected controls (A, n = 5). Arrow: disordered YSL nuclei. YCL, yolk cytoplasmic layer, Bl, blastoderm. Scale bar: 100 μm. (**C**) Distribution of MOs upon injection at the zygote or 1 k-cell stages, as indicated in the schematic. Injection of mondoa-mo together with a lissamine-tagged fluorescent MO of unrelated sequence (tagged-mo) into the yolk of 1 cell stage eggs leads to fluorescence in the entire embryo, whereas injection of the MOs into the yolk cell of the 1 k-cell stage embryos limits fluorescence to the YSL. (**D**) Embryos injected at the 1 k-cell stage with the indicated MOs shown when the uninjected control (n = 29/29, 100%) was at bud stage. All mondoa-mo injected embryos were arrested in epiboly (~75% epiboly stage; n = 9/9, 100%), while embryos injected with tagged-mo alone were unaffected (n = 12/12, 100%). Scale bar: 0.2 mm. Asterisks label the blastoderm margin. hpf, hours post fertilization. The online version of this article includes the following figure supplement(s) for figure 4:

**Figure supplement 1.** EVL epiboly and integrity, endocytic vesicle formation at the marginal rim and overall blastoderm patterning are not severely perturbed in *mondoa* morphants.

domain. Embryos homozygous for this allele did not show an aberrant embryonic phenotype and were raised to sexual maturity (*Figure 7B,C*). By contrast, we found a few maternal-zygotic (MZ) mutants from incrosses of homozygous mutant parents that showed a severe epiboly delay and an arrest in mid-epiboly (*Figure 7B,C*).

The low penetrance of the epiboly phenotype in the MZ mutants is indicative of compensatory mechanisms that counteract the loss of function of *mondoa*. To test this assumption, we injected mondoa-mo into *MZmondoa* mutants. Indeed, we observed a resistance of the mutants to the perturbance of epiboly caused by this MO in wild-types (*Figure 7D*). This observation strongly suggests the presence of strong compensatory or buffering mechanisms that enable embryonic development even when MondoA function is genetically perturbed. It also further confirms the specificity of the morphant phenotype.

## MZ*mondoa* mutants show compensatory changes in expression of cholesterol/steroid biosynthesis genes

To begin to explore potential compensatory mechanisms allowing epiboly progression in MZ*mondoa* mutants, we performed total RNA sequencing of MZ*mondoa* embryos with severe and no epiboly anomalies and of wild-type embryos to determine differential gene expression (*Supplementary file 1*). To compare the gene expression signatures of MZ*mondoa* mutants with and without aberrant epiboly phenotype to those of *mondoa* morphants and their respective controls, we employed the rank-rank hypergeometric overlap (RRHO) algorithm (*Plaisier et al., 2010*). The analysis revealed that differential gene expression significantly overlapped between mutants and morphants, demonstrating that both conditions reflect a lack of MondoA function. Specifically, mutant embryos showed an overlap in differential gene expression with respect to the morphants mainly in the upregulated gene fraction. There were also some differences apparent between the mutants with and without an aberrant epiboly phenotype. We observed that the overlap in downregulated genes was weaker between morphants and mutants without aberrant epiboly phenotype (compare lower left quadrant in *Figure 7—figure supplement 2B and C*). This observation indicated that mainly the downregulated genes in *mondoa* morphants were transcriptionally compensated in MZ*mondoa* mutants with unaffected epiboly. In contrast, the affected mutants showed

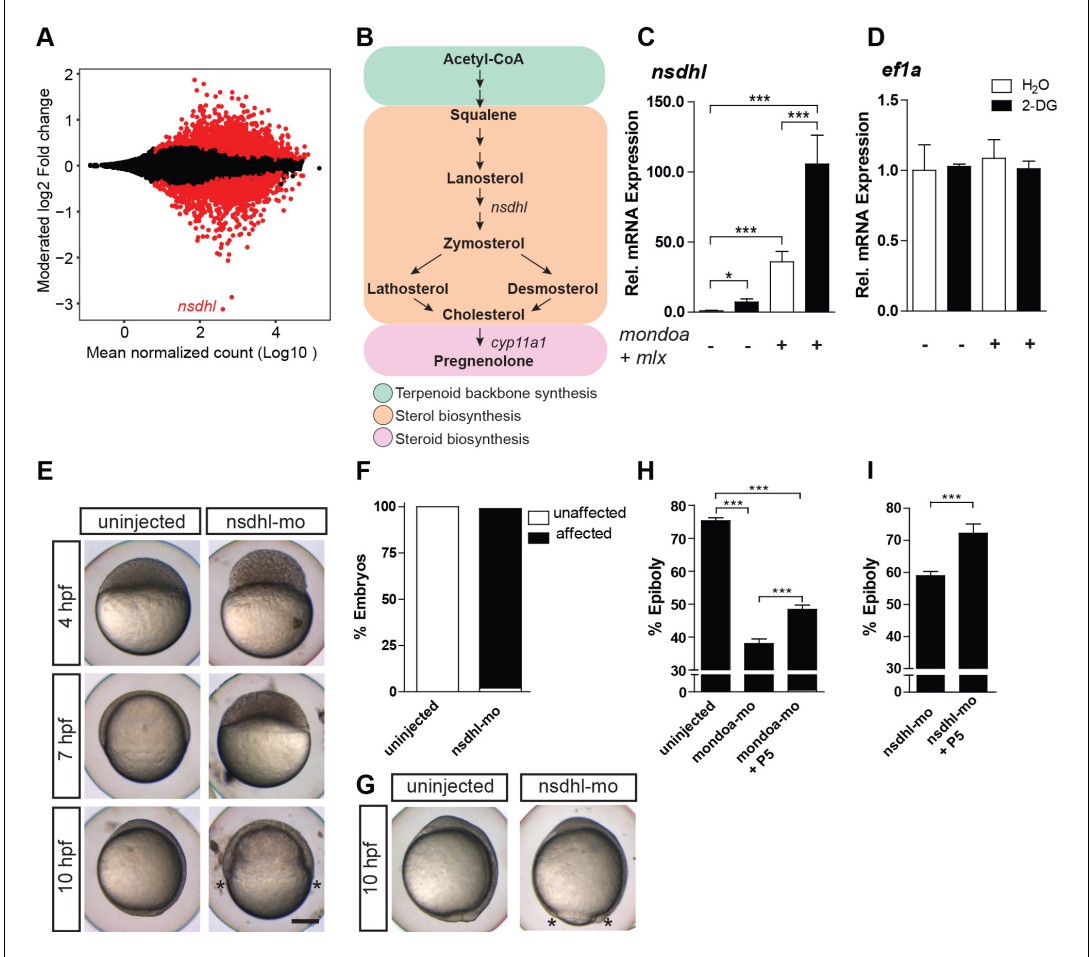

**Figure 5.** MondoA function mediates epiboly via the cholesterol synthesis pathway. (A) Plot of the moderated log2 fold change (mondoa-mo *vs.* mondoa-mis injected embryos) over averaged normalized counts. The most downregulated gene *nsdhl* is indicated. (B) The main steps in the conversion of acetyl coenzyme A (acetyl-CoA) to cholesterol and pregnenolone. (C, D) Transcript levels of *nsdhl* (C) and *ef1a* (D) of embryos injected with water (white bars) as a control or with the glucose analog 2-deoxy-D-glucose (2-DG; black bars) in the presence (+) or absence (-) of *mondoa* and *mlx* mRNA (n = 9). (E) Embryos injected with a splice-site MO against *nsdhl* (nsdhl-mo) show a severe developmental delay and arrest at ~50% epiboly, when uninjected controls were at bud stage (~10 hpf). (F) Quantification of the experiments in (E). Percentages of 'affected' embryos: uninjected (0/84, 0%), nsdhl-mo injected (n = 136/141, 96.5%). (G) Injection of nsdhl-mo into 1 k cell stage embryos led to arrest at 95% epiboly (n = 11/11, 100%), when uninjected control embryos (n = 15/15, 100%) were at bud stage (10 hpf). (H,I) Percentage of epiboly progression under different treatments determined via *no tail* (*ntl*) WISH. (H) Uninjected control embryos (n = 30) were fixed at the 75% epiboly stage along with the treated embryos. Treatment with 20 µM pregnenolone (P5) led to a partial rescue of the *mondoa* morphants (n = 19), which achieved 48% epiboly (untreated morphants: 38% epiboly; n = 25). Control and mondoa-mo morphant results are reproduced from *Figure 2E*, as these experiments were carried out in parallel. (I) P5 treatment also led to a partial rescue of the *nsdhl* morphants (n = 30, 74.6% epiboly compared to 59.0% in untreated morphants, n = 50). Asterisks label the blastoderm margin. Scale bar: 0.2 mm. Error bars represent SEM; *, p≤0.05; **, p≤0.01; ***, p≤0.001.

The online version of this article includes the following figure supplement(s) for figure 5:

**Figure supplement 1.** RNA-seq analysis of mondoa-mo *vs.* mondoa-mis injected embryos identifies nsdhl as a major MondoA target gene.

a weaker overlap in the upregulated gene signatures, suggesting that only a fraction of the upregulated genes in MZ*mondoa* mutants was actually important for phenotype compensation. We next explored these differential gene expression patterns in more detail.

As paralogues of genes have been implicated in compensation (*El-Brolosy et al., 2019*; *Rossi et al., 2015*), we looked at expression of other Mondo pathway components. Neither *chrebp*, the paralogue of *mondoa*, nor the MondoA partner *mlx* were significantly upregulated in MZ*mondoa* mutants, even though trends (p=0.06) for elevated mRNA levels were seen for *mlx* in both mutant phenotypes and for *chrebp* in the affected phenotype only (*Figure 7E*).

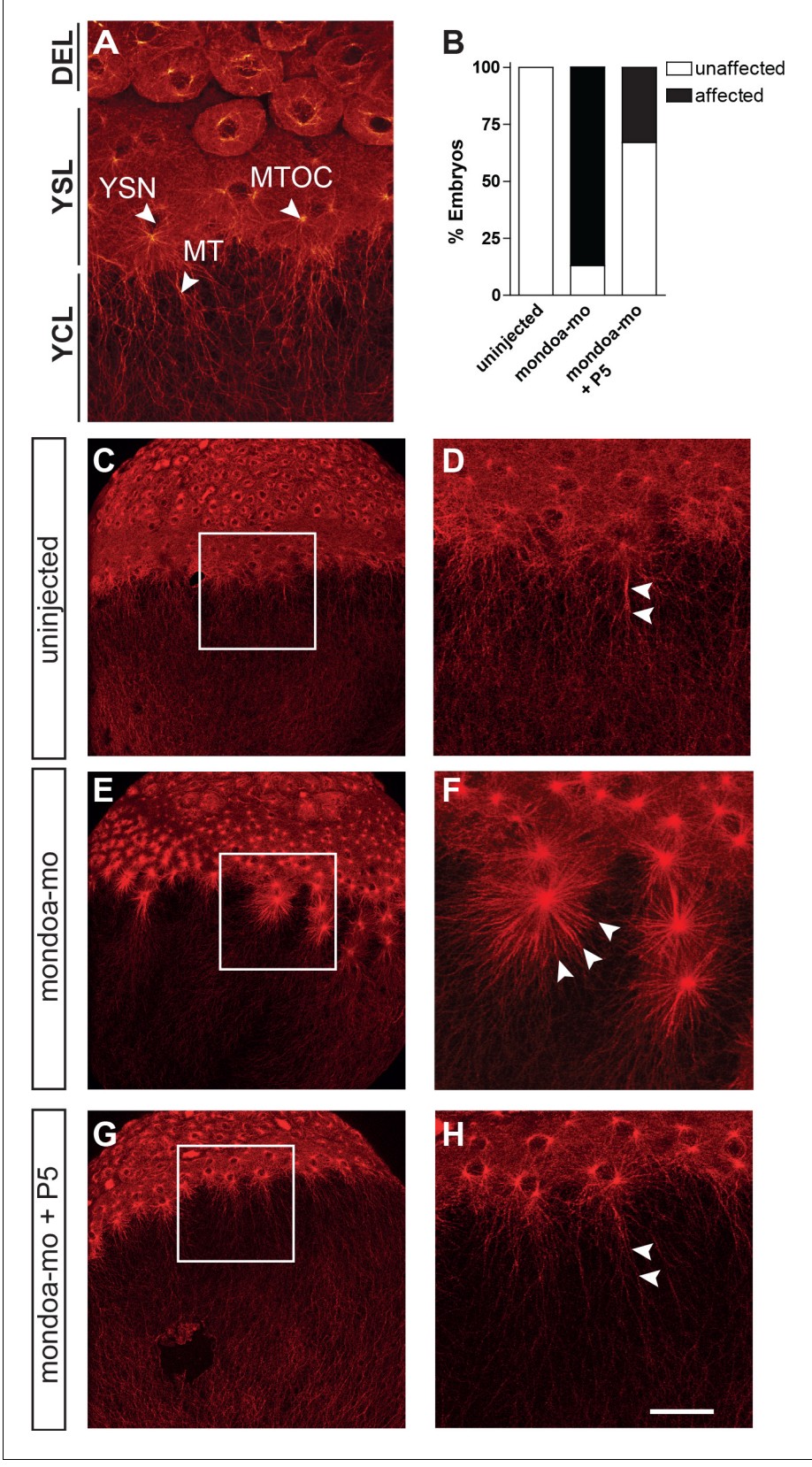

**Figure 6.** Knockdown of *mondoa* affects microtubule stability. (**A**) Confocal image of the blastoderm margin of an uninjected embryo at sphere stage; microtubules are stained with a mouse α-tubulin antibody followed by an anti-
*Figure 6 continued on next page*

*Figure 6 continued*

mouse Alexa Fluor 488-coupled antibody. DEL; deep cell layer; MT, microtubules; MTOC, microtubule organization center; YCL, yolk cytoplasmic layer; YSL, yolk syncytial layer; YSN, yolk syncytial layer nuclei. (**B**) Quantification of effects on YSL microtubule organization in uninjected (n = 9), mondoa-mo injected (n = 16), and mondoa-mo injected + pregnenolone (P5) treated (n = 9) embryos. Presence of stellar structures as shown in (**F**) = 'affected', normal appearance as in (**A**) = 'unaffected'. (**C–H**) Overview (**C, E, G**) and close-up (**D, F, H**) images of uninjected embryos (**C, D**), mondoa-mo morphants (**E, F**) and P5 treated mondoa-mo morphants (**G, H**). Arrowheads indicate long parallel MT in uninjected and P5 rescued embryos (**D, H**) and short MT asters in untreated morphants (**F**). Scale bar: 90 μm; for higher magnifications, 30 μm.

We next turned our attention to genes in the cholesterol biosynthesis pathway based on our previous observations in morphant embryos. Our analysis showed that *nsdhl* gene expression was comparable to wild-type embryos in both MZ*mondoa* mutant phenotypes (affected and unaffected epiboly) and, thus, was compensated in both (***Supplementary file 1*** and ***Figure 7—figure supplement 2E–F***). Therefore, differences in *nsdhl* expression between MZ*mondoa* mutants cannot explain the differences in the epiboly phenotype. This observation indicates that the restoration of expression to wild-type levels of this most downregulated gene upon *mondoa* knockdown by itself was not sufficient for compensation. Consistently, in both mutant phenotypes the entire sterol biosynthesis pathway of which *nsdhl* forms part (***Figure 5B***) lacked the coordinated downregulation that was seen in the morphants (unaffected: p=0.87, affected: p=0.79; ***Figure 7F*** and ***Figure 7—figure supplement 1A***).

But what is causing the epiboly arrest if MZ*mondoa* mutants with affected epiboly share such compensatory transcriptional signatures with MZ*mondoa* mutants without aberrant epiboly phenotype? *mondoa* morphants showed a significant downregulation in terpenoid backbone biosynthesis (p=0.016) as well as in sterol biosynthesis pathways (p=0.022; ***Supplementary file 2***). By contrast, MZ*mondoa* mutants with unaffected epiboly showed a coordinated upregulation of terpenoid pathway genes (i.e., they showed a statistically significant enrichment for upregulation in a directional gene set enrichment test; p=0.012; ***Figure 7F–G***, ***Figure 7—figure supplement 1B*** and ***Supplementary file 2***). Intriguingly, mutants with affected epiboly lacked this significant coordinated upregulation in the terpenoid backbone biosynthesis pathway that was present in unaffected mutants (p=0.178; ***Figure 7F–G*** and ***Figure 7—figure supplement 1B***). This key difference highlights the upregulation of gene expression in the terpenoid backbone biosynthesis pathway as a compensation response specifically required to restore the normal epiboly phenotype.

We additionally observed that gene expression of steroid hormone biosynthesis pathway genes showed a coordinated upregulation in MZ*mondoa* mutants with unaffected epiboly (p=0.033; ***Figure 7F*** and ***Figure 7—figure supplement 1C***), probably to compensate for reduced flux through the biosynthesis pathways upstream. This upregulation was also shown by MZ*mondoa* mutants with aberrant epiboly phenotype (p=0.017; ***Figure 7F*** and ***Figure 7—figure supplement 1C***). Strikingly, it was even increased when compared to the mutants with unaffected epiboly (p=0.049), and thereby led to strongly increased expression levels compared to wild-type. This is illustrated by the expression of *cyp11a1*, the enzyme that is immediately upstream of pregnenolone synthesis (***Supplementary file 2***, ***Figure 7—figure supplement 2G***). However, in the absence of coordinated regulation of genes in the terpenoid pathway, this regulation was apparently not sufficient to rescue epiboly progression.

Taken together, despite some successful compensatory gene expression regulations, MZ*mondoa* mutants with affected epiboly fail to properly regulate the entire set of genes necessary for epiboly progression. They are not lacking compensation per se, but their compensation is inadequate and fails.

In summary, results from both MO mediated and genetic loss-of-function of MondoA revealed MondoA as a key developmental regulator of progression through epiboly by regulating expression of cholesterol/steroidogenesis pathway genes important for yolk cell microtubule function.

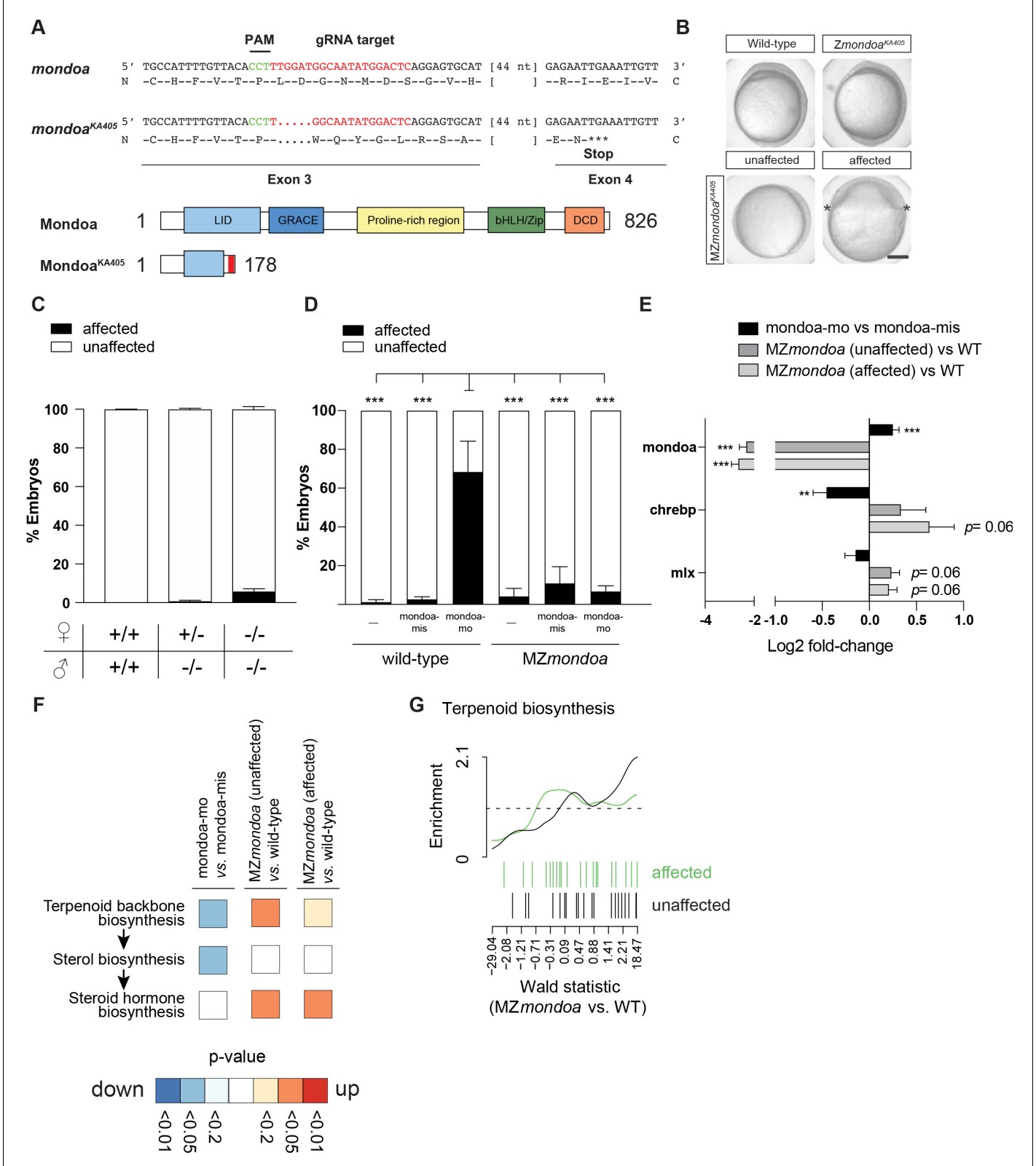

**Figure 7.** MZ*mondoa* mutants show epiboly defects and compensatory gene expression changes in the terpenoid backbone biosynthesis pathway. (A) Genetic disruption of the *mondoa* (*mlxip*) locus in zebrafish led to a 5 bp deletion in exon three causing a frameshift in the coding sequence and a predicted premature stop. (B) Representative images of zygotic homozygous mutants (Z*mondoa*^Ka405^) and maternal-zygotic mutants (MZ*mondoa*^Ka405^). (C) Quantification of epiboly phenotypes. On average 5.09 ± 1.25% of MZ*mondoa* mutants (n = 249/5119 embryos) showed an aberrant epiboly

*Figure 7 continued on next page*

*Figure 7 continued*

phenotype compared to 0.10 ± 0.07% in wild-type (WT; n = 1/1016 embryos) and 0.71 ± 0.39% in Z*mondoa* mutant embryos (n = 3/549 embryos). (D) MZ*mondoa* mutants exhibited resistance to epiboly perturbance caused by mondoa-mo, confirming specificity of the morphant phenotype (n = 3, ≥18 embryos/replicate and condition, treatment/genotype combinations blinded for analysis). (E–H) RNA-seq gene expression analysis in mondoa-mo *vs.* mondoa-mis injected embryos (n = 3, ≥20 embryos/replicate) and MZ*mondoa* mutants *vs.* WT embryos (n = 3, 8–10 embryos/replicate). (E) Differential gene expression of *mondoa*, *chrebp* and *mlx* in morphants and MZ*mondoa* mutants with (affected) and without (unaffected) aberrant epiboly phenotype. (F) Heatmap illustrating the statistical overrepresentation of pathway associations for morphant (mondoa-mo *vs.* mondoa-mis) or mutant embryos (MZ*mondoa* *vs.* WT) affected or unaffected in epiboly. (G). Overlaid barcode plots illustrating the enrichment for differential gene expression compared to controls for terpenoid backbone biosynthesis genes. MZ *mondoa* mutants with unaffected epiboly (black) display a coordinated upregulation of genes associated with terpenoid biosynthesis. MZ*mondoa* mutants that are affected in epiboly (green) lack this coordinated upregulation. Error bars represent SEM; *, p≤0.05; **, p≤0.01; ***, p≤0.001.

The online version of this article includes the following figure supplement(s) for figure 7:

**Figure supplement 1.** Gene set enrichment analysis of cholesterol related pathways in *mondoa* morphants and mutants.

**Figure supplement 2.** Overlaps and differences between *mondoa* morphants and mutants in genes differentially expressed compared to controls.

## Discussion

Here, we have characterized the Mondo pathway in zebrafish and revealed a developmental function for Mondo signaling, implying a novel role for this glucose-sensing pathway in addition to its functions in adult animals (*Song et al., 2019*). Previously, studies in non-vertebrates have implicated homologs of Mondo pathway factors in later stages of embryonic or larval development: Loss of function of the Mlx orthologue in *Drosophila* is lethal at the late pupal stage (*Havula et al., 2013*), while knockdown of a distant Mlx homolog in *C. elegans* affected movements of ray one precursor cells (*Pickett et al., 2007*), which will form a subset of male-specific sense organs in the tail. To the best of our knowledge, ours is the first study demonstrating conserved Mondo signaling mechanisms between zebrafish and mammals and linking its function to early embryonic development of vertebrates.

Our findings demonstrate that MondoA acts upstream of the synthesis of pregnenolone, an important regulator of yolk cell microtubule stability and epiboly movements (*Hsu et al., 2006*; *Schwend et al., 2011*; *Weng et al., 2013*), by regulating the expression of a whole set of genes in the terpenoid/sterol biosynthesis pathway, as exemplified by the cholesterol synthesis enzyme *nsdhl*. It is tempting to speculate that the role of MondoA in cholesterol/steroid metabolism we here uncover in the zebrafish is a feature conserved across vertebrates, albeit not much is known in mammals. One *in vitro* study described 'sterol biosynthetic process' as one category of genes upregulated in MondoA deficient HeLa cells under conditions of acidosis (*Wilde et al., 2019*). For the MondoA paralogue Chrebp, a study *in vivo* reported that 'cholesterol biosynthesis' genes are upregulated in the liver of *Chrebp* knockout mice which were fed with a high fructose diet (*Zhang et al., 2017*). While these studies were performed under specific disease conditions (i.e., a human cancer cell line under acidosis conditions or animals fed with a high fructose diet), they support our observations that the Mondo pathway plays an important role in the regulation of cholesterol/steroid metabolism also under non-disease conditions. Furthermore, these observations indicate gene expression changes due to compensatory mechanisms upon loss of Mondo pathway function, as observed in our study and discussed in the following section.

The low penetrance of epiboly phenotypes in MZ*mondoa* mutant zebrafish embryos indicates the presence of efficient compensatory pathways, triggered by the genetic lesion but not by antisense-mediated knockdown, that may be operating in mammals as well (*El-Brolosy and Stainier, 2017*). Nonsense mediated decay products of the mutated transcript are proposed to trigger transcriptional upregulation of homologous genes that compensate for the mutant gene's function (*El-Brolosy et al., 2019*). However, *chrebp* and *mlx* mRNA levels were not significantly changed compared to wild-type in mutants with unaffected epiboly, implicating the presence of alternative compensatory mechanisms. Interestingly, many terpenoid, sterol and steroid biogenesis genes showed lower than wild-type levels of expression in the morphants and equal or higher levels in the mutants with unaffected epiboly, suggesting that increased steroidogenesis activity rescues epiboly in a

majority of MZ mutant embryos. In the remaining mutant embryos, the lack of coordinated upregulation of terpenoid backbone biosynthesis genes is indicative of an incomplete compensation attempt. Apparently, concerted changes across the entire cholesterol/steroid biosynthesis pathway are needed to rescue epiboly progression in the absence of MondoA function. The mere restoration of *nsdhl* expression levels, for example, appears not to be sufficient for such a rescue, perhaps because a few genes in the sterol biosynthesis pathway still show deregulation of expression in MZ mutant embryos of both phenotypes (*Figure 7—figure supplement 1A*). Therefore, even though globally there is no coherent up- or downregulation in sterol biosynthesis gene expression of mutants compared to wild-type embryos based on gene set enrichment analysis, pathway function in mutants may still be suboptimal and require the compensatory upregulation in the upstream terpenoid pathway to achieve sufficient synthesis flux. Furthermore, random variation in the amount of material deposited by the mother or in protein expression of key genes may limit compensation in the mutants with affected epiboly. A more detailed understanding of these processes awaits the determination of metabolite fluxes in morphants, mutants and wild-type embryos, specifically in the YSL. Importantly, incomplete penetrance is a frequent phenomenon in human genetic diseases and has been attributed to a wealth of mechanisms, which have been suggested as targets for therapy (*Cooper et al., 2013*). Future studies how disturbed MondoA function is buffered during development might open leads to manipulating dysregulated Mondo pathway function in adult metabolic regulation.

We observed that *mondoa* knockdown in the YSL was sufficient to cause an epiboly phenotype, pointing towards a crucial role of MondoA in this tissue. Even though the YSL and epiboly movements are specific for teleosts, there are several indications that the mechanisms detected in the zebrafish are relevant in mammals as well. For example, steroidogenic enzymes are expressed in extraembryonic tissues in both fish and mice, the fish YSL (*Hsu et al., 2006*; *Hsu et al., 2009*) and the giant trophoblast cells of the murine placenta (*Arensburg et al., 1999*). Indeed, it has been proposed that zebrafish epiboly is analogous to the spreading of the trophectoderm during implantation (*Kane and Adams, 2002*). Consistent with this idea, epiboly movements are severely impaired upon knockdown of the zebrafish homolog of solute carrier family 3 member 2 (*Slc3a2*) (*Takesono et al., 2012*), a factor crucial for trophoblast cell adhesion, migration and fusion in mammals (*Kabir-Salmani et al., 2008*; *Kudo et al., 2003*). Thus, it is conceivable that developmental functions of MondoA are conserved among vertebrates. Testing this hypothesis awaits a close examination of early embryonic and placental phenotypes in MondoA mutant mice, the analysis of which has so far been limited to adult phenotypes (*Ahn et al., 2019*; *Imamura et al., 2014*), and of potential functions of maternal MondoA mRNA or protein in mammals (*Li et al., 2010a*). Given the small litter size of rodents, phenotypes with low penetrance may easily be overlooked and are difficult to study in these animals.

Importantly, mutations in the *NSDHL* gene in humans have been linked to an X-linked dominant condition called CHILD syndrome ('congenital hemidysplasia with ichthyosiform erythroderma and limb defects', reviewed in *Herman, 2000*) that is usually lethal in males (*Bornholdt et al., 2005*). Murine male embryos carrying *Nsdhl* mutations die shortly after implantation, and those carrying moderate or mild alleles showed reduced placental thickness with a poorly vascularized fetal labyrinth (*Caldas et al., 2005*; *Liu et al., 1999* and references therein). Interestingly, also in female mouse embryos carrying a maternal null allele placental area was severely reduced (*Cunningham et al., 2010*). In these embryos, paternal X chromosome inactivation in yolk sac endoderm and trophoblast-derived lineages causes complete loss of function only in these tissues. Given the potential analogous functions of placenta and YSL (*Carvalho and Heisenberg, 2010*), our results regarding Nsdhl function in epiboly make it tempting to speculate that Nsdhl contributes to the development of the mammalian placenta by regulating microtubule stability. Remarkably, stability of microtubules apparently plays a role in trophoblast differentiation and implantation based on *in vitro* studies (*Bates and Kidder, 1984*; *Douglas and King, 1993*).

In summary, our results identify a novel role for the Mondo pathway in vertebrate development and link glucose sensing to cholesterol/steroid biogenesis. These results contribute to the growing recognition of metabolic regulatory functions in development (*Miyazawa and Aulehla, 2018*) and broaden our understanding of Mondo signaling.

# Materials and methods

**Key resources table**

| Reagent type (species) or resource | Designation | Source or reference | Identifiers | Additional information |
|---|---|---|---|---|
| Gene (*Danio rerio*) | mlxipl | This paper | KF713494 | Sequence deposited at https://www.ncbi.nlm.nih.gov/nuccore |
| Gene (*Danio rerio*) | mlxip | This paper | KF713493 | Sequence deposited at https://www.ncbi.nlm.nih.gov/nuccore |
| Genetic reagent (*Danio rerio*) | Mlxip mutant (allele: ka405) | This paper | ZFIN ID: ZDB-ALT-180628–2 | Detailed info deposited at zfin.org |
| Cell line (*Danio rerio*) | PAC2 | *Lin et al., 1994* | | |
| Cell line (*Homo sapiens*) | HepG2 | ATCC | Hep G2 [HEPG2] ATCC HB-8065 | |
| Sequence-based reagent | mondoa-mo | This paper | Morpholino | 5'-ggtattgtcgagt agccatgttaaa-3' |
| Sequence-based reagent | mondoa-mis | This paper | Morpholino | 5'-ggtaatctccag tacccatcttaaa-3' |
| Sequence-based reagent | nsdhl-mo | This paper | Morpholino | 5'- ctgaaacaattc acacctgtttgtc-3' |
| Sequence-based reagent | p53-mo | *Robu et al., 2007* | Morpholino | 5'-gcgccattgct ttgcaagaattg-3' |
| Sequence-based reagent | mlx-mo | This paper | Morpholino | 5'-cgcgctgttttcc gtcattttggaa-3' |
| Sequence-based reagent | chrebp-mo | This paper | Morpholino | 5'-ttctgaatactct gcttttgccatc-3' |
| Sequence-based reagent | tagged-mo | This paper | Morpholino | 5'-gtagcttgtactc gcattcctatct-3' |
| Recombinant DNA reagent | Plasmid constructs | This paper | Plasmid | Plasmid constructs listed in "Materials and methods" and generated for this study are available upon request from the Dickmeis laboratory |

## Fish stocks and maintenance

All zebrafish husbandry was performed in accordance with the German animal protection standards and approved by the Government of Baden-Württemberg, Regierungspräsidium Karlsruhe, Germany (Aktenzeichen35-9185.64/BH KIT), by the Home Office, United Kingdom (Scientific Procedures, Act 1986), and by the Service de la consommation et des affaires vétérinaires du canton de Vaud, Switzerland. Wild-type fish were descendants of the AB strain (University of Oregon, Eugene) or the golden strain and have been reared for several years in the laboratory. Fish were bred and raised in E3 medium (*Nüsslein-Volhard, 2002*) (5 mM NaCl, 0.17 mM KCl, 0.33 mM CaCl$_2$, 0.33 mM MgSO$_4$,

0.1% methylene blue). Staging of embryos was performed on a dissecting microscope according to *Kimmel et al., 1995*.

## Cloning of cDNAs and other plasmid constructs

### Cloning of cDNAs

In order to obtain a full length cDNA sequence for *chrebp*, we performed a PCR with the primer pairs PCR_chrebp_1 (fw: 5'-aaccgcagaaccgaatcag-3', rv: 5'-tagctgtgcgatgaatgtcc-3') on a commercially available cDNA template (Life Technologies, 1 cell stage). 5' and 3' ends of the cDNA were determined with 5' and 3' RACE using the FirstChoice RLM-RACE Kit as recommended by the manufacturer (Ambion). A 5' RACE was also performed to confirm the translation start site of *mondoa*.

The open reading frames (ORFs) of *mondoa* and *mlx* were cloned by nested PCR using Platinum Taq DNA polymerase as recommended by the manufacturer (Life Technologies). The first PCR was carried out with the primers PCR_mondoa_1 (fw: 5'-ggacgaaggaattgacgaaa-3', rv: 5'-ctaagaccctgaggaatgtt-3') and PCR_mlx_1 (fw: 5'-attgtttattccaaaatgac-3', rv: 5'-gtgctgagccctgctgtatt-3') for *mondoa* and *mlx*, respectively. The cDNA template used was a commercially available cDNA (Life Technologies, 1 cell and 18–30-somite stages). PCR products obtained were used as templates for the second PCR round with the primers PCR_mondoa_2 (fw: 5'-atggctactcgacaataccgga-3', rv: 5'-ccctgaggaatgttttttgggg-3') and PCR_mlx_2 (fw: 5'-atgacggaaaacagcgcgt-3', rv: 5'-cagttgaccatt-cacctgctgc-3') for *mondoa* and *mlx*. The final PCR products were subcloned into the pGEMT-Easy vector and sequenced.

### Expression vectors for *mondoa* and *mlx*

For subcloning into an expression vector to perform overexpression studies, a further PCR step was carried out with the Platinum Taq DNA polymerase kit, using pGEMT-Easy:MondoA and pGEMT-Easy:Mlx as templates and the primers PCR_mondoa_3 (fw: 5'-taggtacccatggctactcgacaataccgga-3', rv: 5'-tagctcgagccctgaggaatgttttttgggg-3') and PCR_mlx_3 (fw: 5'-taggtacccatgacggaaaacagcgcgt-3', rv: 5'-tagctcgagcagttgaccattcacctgctgc-3'), which include XhoI and KpnI overhangs. The PCR products were cloned into the XhoI and KpnI restriction sites of a pcDNA3.1(–) vector (Life Technologies) which lacked the neomycin cassette (removed with an NaeI digest), resulting in the pcDNA3.1(–)ΔNeo:MondoA and pcDNA3.1(–)ΔNeo:Mlx vectors, respectively. For the *mondoa* mRNA rescue experiments, silent mutations were created in the pcDNA3.1(–) ΔNeo:MondoA vector so that the protein sequence was retained, while mondoa-mo binding was abolished. Mutagenesis was performed with the QuikChange site-directed mutagenesis kit as described by the manufacturer (Stratagene). The primer used was PCR_mondoa_res (5'-ggctagttaagcttggtacccatggcaacgaggcagtataggagggccacaatgatgatcaagca-3'). RNA was transcribed using the mMESSAGE mMACHINE Kit (T7) (Ambion) on templates linearized with DraIII.

### Luciferase reporter constructs

The constitutively expressed pGL3-Control luciferase vector was obtained from Promega (#E1741). The PathDetect pGRE-Luc cis-Reporter plasmid was obtained from Agilent Technologies (#240133). The reporter constructs pLucMCS:2xChoRE and pT2Luci:2xChoRE were generated by cloning the following annealed 5' phosphate modified oligonucleotides into the XhoI site of the pLucMCS and pT2Luci:MCS recipient plasmids, respectively, as previously described (*Weger et al., 2011*; *Weger et al., 2012*; *Weger et al., 2013b*): 2xChoRE oligos: fw: 5'-tcgatcacgagtgggccgcgtgccggc-cacgagtgggccgcgtga-3'; rv: 5'-tcgatcacgcggcccactcgtggccggcacgcggcccactcgtga-3'.

### pCS2:GFP:MondoA and pCS2:GFP:ChREBP knockdown efficiency constructs

To test the knockdown efficiency of the mondoa-mo, the ATG region of *mondoa* was fused to a GFP reporter gene. The ATG region of *mondoa* (+36/–20) was generated by annealing of the following oligonucleotides containing NcoI and EcoRI overhangs: MondoA_1: fw: 5'-aattatacttatagtaaaagtc-gaagcg-3', rv: 5'-gacgcgcttcgactttactataagtat-3'; MondoA_2: fw: 5'-cgtattttaacatggctactcgacaa-taccg-3', rv: 5'-catgcggtattgtcgagtagccatgttaaaaa-3'. These oligonucleotides were then inserted in frame in front of the GFP of a pCS2:GFP vector (*Ertzer et al., 2007*) linearized with NcoI and EcoRI, yielding the pCS2:GFP:MondoA construct. This construct was linearized with NotI and transcribed

with the mMESSAGE mMACHINE (Sp6) kit as recommended by the manufacturer (Ambion). The ATG region of *chrebp* (+26/–34) was generated similarly by annealing the following oligos: chrebp_1: fw: 5'-aattgcagtttgtgaagagtgaataacactag-3', rv: 5'-agaactagtgttattcactcttcacaaactgc-3'; chrebp_2: fw: 5'-ttctcttgcgatggcaaaagcagagtattcagaaac-3', rv: 5'-catggtttctgaatactctgcttttgccatcg-caag-3'.

## Templates for digoxigenin labeled probes

In order to generate the templates for WISH probes directed against the *chrebp*, *mondoa*, *mlx* and *nsdhl*, PCR was carried out with the following oligos: *chrebp* probe 1: fw: 5'-ttcatctgcgctcaacactc-3', rv: 5'-ttgagatctggacgagagca-3'; *chrebp* probe 2: 5'-ctacagccagagggaagcac-3', rv: 5'-gatccgtcgtgtgtctgttc-3'; *mondoa* probe 1: fw: 5'-actaccgtcagccacaggtc-3', rv: 5'-cactctgcgactct-gactgtt-3'; *mondoa* probe 2: fw: 5'-ttccttctctgcctctggac-3', rv: 5'-ccagcagaaccacaaggaat-3'; *mlx* probe 1: fw: 5'-gatggcaccttcagtgacaa-3', rv: 5'-tggcttgcagtgttcctcta-3'; *mlx* probe 2: fw: 5'-tccacact-taggaaggaggtg-3', rv: 5'-tgattggaatcactgccata-3'; *nsdhl* probe 1: fw: 5'-agcgcttagtacaccgagga-3', rv: 5'-cgagcgcagctgtagtaatg-3'; *nsdhl* probe 2: fw: 5'-tcatacaggcttgccatgag-3', rv: 5'-tcctaggctcgtcttgcagt-3'. PCR products were subcloned into the pGEMT-Easy vector (Promega). The *no tail* (*ntl*), *goosecoid* (*gsc*), *ventral homeobox* (*vox*), *ventral expressed homeobox* (*vent*) and *casanova/SRY box containing gene 32* (*sox32*) probes were isolated in a large scale transcription factor screen (**Armant et al., 2013**). Digoxigenin labeled probe synthesis was done following the manufacturer's instructions (DIG RNA labeling mix, Roche).

## RNA extraction, cDNA Synthesis, RT-qPCR

MZ*mondoa* and wild-type RNA was extracted using RNA Clean and Concentrator-25 columns with an included DNase I treatment (ZYMO RESEARCH) for the corresponding RNA-seq experiment, as described in the manufacturer's protocol. RNA that was used for cDNA synthesis was extracted from cells and whole embryos with the TRIzol reagent (Life Technologies), as well as RNA that was prepared for the RNA-seq experiment of MondoA morphants. TRIzol extraction was performed as described in the manufacturer's protocol, with minor modifications. If not otherwise stated, 30 embryos were harvested in 1.5 ml TRIzol. Embryos were disrupted in the TRIzol solution with micropestles (Eppendorf), and $5 \times 10^5$ cells in 25 cm$^2$ flasks were harvested with 1.5 ml TRIzol solution. Samples were then stored overnight at –80°C. After thawing the samples on ice, they were passed through a syringe four times (Braun Sterican, $0.45 \times 25$ mm). The integrity of the total RNA was checked on an agarose gel. A NanoDrop spectrometer was used to determine concentrations and A260/A280 ratios of the samples. Only RNA with an A260/A280 ratio above 1.6 was used. cDNA was synthesized using random primers and Superscript III reverse transcriptase according to the manufacturer's protocol (Life Technologies). Transcript levels of genes of interest were determined with the StepOnePlus Real-Time PCR System (Applied Biosystems) using the QuantiTect SYBR Green PCR Kit (Qiagen). Copy numbers were normalized using *β-actin* transcript levels. The melting curves of the amplicons were determined after amplification. Ct values were determined using StepOne Software 2.1 (Applied Biosystems) and data were analyzed by the $2^{-\Delta\Delta Ct}$ method as previously reported (**Livak and Schmittgen, 2001**). Primer sequences: *β-actin*: fw: 5'-gcctgacggacaggtcat-3', rv: 5'-accgcaagattccataccc-3'; *eef1a*: fw: 5'-ccttcgtccaatttcagg-3', rv: 5'-ccttgaaccacccccatgt-3'; *mlxip*: fw: 5'-ctgccagacgtatcacttcg-3', rv: 5'-ggcgaatcttgtctctccac-3'; *mlxipl*: fw: 5'-ccacagccagtgttgatgat-3', rv: 5'-tagctgtgcgatgaatgtcc-3'; *mlx*: fw: 5'-agacttctccatggccacac-3', rv: 5'-ctgatgggccttcacaatg-3'; *pck1*: fw: 5'-tgacgtcctggaagaacca-3', rv: 5'-gcgtacagaagcgggagtt-3'; *hk2*: fw: 5'-gaaattggcct-cattgtcg-3', rv: 5'-catccaccagctccaagtg-3'; *fasn*: fw: 5'-aggccatcgttgatggag-3', rv: 5'-tgtagacgc-cagttttgctg-3'; *txnipa*: fw: 5'-ccaactagacgaacatccaaca-3', rv: 5'-agacaccagctgcccttg-3'; *sox3*: fw: 5'-cttagcgcacaactttgcag-3', rv: 5'-caccagtcccgtgtgtctc-3'; *tmprss4b*: fw: 5'-tcaaagtttcctctcagccagt-3', rv: 5'-ccaccacctcaccacagtc-3'; *adh5*: fw: 5'-cacgctcctctggataaagtgt-3', rv: 5'-gtagagcccgcttcaacct-3'; *cfl1*: fw: 5'-atgatctacgccagctccaa-3', rv: 5'-tcacttgccactcgtgctta-3'; *vent*: fw: 5'-aggagaaatgcagca-cagc-3', rv: 5'-tcactctccacatcggtgtatc-3'; *lmnl3*: fw: 5'-cctggccaactacatcgag-3', rv: 5'-gatctgcatggag-gatttgtc-3'; *dynlrb1*: fw: 5'-aacagcggtttgtcttcaca-3', rv: 5'-atggggattccttctgcatt-3'; *gabarapl2*: fw: 5'-cgacaaaactgtccctcagtc-3', rv: 5'-ccgtcctcgtcttgttcttt-3'; *aurkaip1*: fw: 5'-cgagagtggtttcacgactg-3', rv: 5'-ctggctctgtgtctgcaatg-3'; *nsdhl*: fw: 5'-ggggacctctgtgacaaaca-3', rv: 5'-aggtgaggcacaatgaaaca-3'.

## Microinjection

If not otherwise stated, all injections were performed as follows: Injection mixes were prepared with the desired concentrations of the compound of interest (see below) and 0.1% (w/v) phenol red as an injection control. Each injection mix was injected with borosilicate glass capillary tubes (~0.6 mm diameter) into the yolk of 1 cell stage zebrafish eggs with a gas-driven microinjector (Eppendorf Femtojet express) as described (*Müller et al., 1999*). After the injection, unfertilized or damaged eggs were removed around the high to sphere stages and were not considered for the subsequent analysis.

## Whole-mount *in situ* hybridization (WISH) and Whole-mount immunohistochemistry (WIHC)

WISH was performed on embryos (n $\geq$ 20) as previously described (*Oxtoby and Jowett, 1993*), with minor modifications: The Proteinase K digest step was omitted for early embryonic stages and replaced with four PTw (1xPBS, 0.1% Tween-20) washes, while it was reduced to 6 min for 24 hr old embryos. Epon embedding of embryos was carried out as described (*Westerfield, 2007*). 5 µm thick sections were imaged with a compound microscope (Leica DM5000). Percentage of epiboly progression of the embryos was determined by performing a *ntl* in situ hybridization to label the blastoderm margin and using the following formula as previously described (*Hsu et al., 2006*): Epiboly (%) = (The length between animal pole and blastoderm margin)/(The length between animal pole and vegetal pole).

Microtubule staining of early embryos by WIHC was carried out as recommended by *McMenamin et al., 2003*. Uninjected embryos and mondoa-mo injected embryos were fixed at late blastula stage just before the onset of epiboly and stained for $\alpha$-tubulin to visualize the microtubules. The primary antibody used was an anti-$\alpha$-tubulin antibody (Sigma-Aldrich, 1:500), the secondary antibody was anti-mouse Alexa Fluor 488 (Life technologies, 1:1000). Phalloidin staining of actin filaments was performed according to *Nakajima and Burke, 1996*. Briefly, embryos were fixed in 4% PFA over night at 4˚C. The next day, fixed embryos were washed three times in PTw and then incubated with Rhodamine-Phalloidin (1:500, R415, Molecular Probes (Life Technologies)) for 15 min. Embryos were mounted in 0.5% (w/v) low melting agarose in E3 and imaged with a confocal microscope (Leica TCS2 SP5). As observed for epiboly movement disruption, the severity of the YSL disorganization appears to be dose-dependent, with lower doses of MO leading to a less severe phenotype.

## Cell culture

### Cell lines and maintenance

Zebrafish PAC2 cells (*Lin et al., 1994*) were maintained essentially as described (*Whitmore et al., 2000*). Briefly, cells were kept in PAC2 maintenance medium (L-15 Medium, 17% (v/v) FBS, 100 U/ml penicillin/streptomycin, 50 mg/ml gentamycin) in 75 cm$^2$ cell culture flasks at 28˚ C and passaged every week by trypsinization at a ratio of 1:7 to 1:10. HepG2 cells (ATCC, #HB-8065) were kept in HepG2 maintenance medium (DMEM, 10% (v/v) FBS, 100 U/ml penicillin/streptomycin) at 37˚C and 5% CO2 in a humidified incubator and were passaged twice a week at the same ratio. Cells were not authenticated by STR profiling. Tests for mycoplasma contamination were performed every 8 weeks, showing negative results.

### Cell transfection

Transient transfections were carried out with FuGENE HD reagent (Promega) following the manufacturer's protocol. Briefly, one day before transfection, cells were washed once with 1x PBS without Ca$^{2+}$ or Mg$^{2+}$, then trypsinized at room temperature to detach cells. To obtain 80% cell confluency on the day of transfection, cell number was determined with a Neubauer chamber using the Trypan blue exclusion method. Cells were diluted with PAC2 maintenance medium and transferred into 6-well plates (800,000 cells per six well), then incubated overnight at 28˚C. The next day, maintenance medium was replaced with maintenance medium without antibiotics. The DNA complex was prepared using 100 µl L-15 medium, 1 µg total DNA of interest and 4 µl FuGENE HD (4:1). The mixture was incubated for 20 min at room temperature to ensure complex formation before the entire reagent mixture was added to the cells. Cells were incubated with the reagent overnight at 28˚C.

The following day, cells were transferred into opaque 96-well plates with PAC2 maintenance medium and were allowed to attach overnight before measurement of bioluminescence (see below). This protocol was modified for the HepG2 cells as follows: 200,000 cells per six wells, DNA complex consisting of 100 µl L-15 medium, 2 µg total DNA of interest and 8 µl FuGENE HD (8:2).

To measure glucose induction of transcription from a ChoRE reporter, cells were transfected either with the 2xChoRE reporter construct or with pGL3-Control (Promega) as a negative control. Treatment of cells with different concentrations of glucose was performed after starvation of HepG2 cells for 1 hr and of PAC2 cells overnight. The starvation medium (i.e. maintenance medium without serum and phenol red) was supplemented with 0.5 mM luciferin. Bioluminescence was determined after 24 hr of glucose incubation (see below).

For overexpression studies with Mondo pathway factors, PAC2 cells were transfected with the 2x ChoRE reporter and co-transfected with full-length *mondoa*, *mlx*, or both. The DNA complex was prepared using 600 ng 2xChoRE reporter DNA, 200 ng pRL-CMV vector (to control for transfection efficiency; Promega #E2261), and a total of 200 ng of transcription factor and carrier DNA (pBluescript SK). For the overexpression of one transcription factor, 100 ng transcription factor and 100 ng carrier DNA were used. For the overexpression of two transcription factors, 100 ng of each transcription factor were employed. The zebrafish nr3c1 (GRα) construct (*Schaaf et al., 2008*) was kindly provided by M. Schaaf (Institute of Biology Leiden, Leiden University, Netherlands). After transfection, cells were starved and then treated for 24 hr with either low (0.3 mM) or high (12 mM) glucose concentrations before bioluminescence was determined. Bioluminescence was measured in cell lysates and normalized to Renilla luciferase activity to control for transfection efficiency.

### Morpholino oligonucleotide mediated knockdown in PAC2 cells

Transient transfection of MOs was done by electroporation essentially as previously described (*Weger et al., 2011*). Briefly, cells (80% confluent 1/2 flask 175 cm$^2$ per transfection) were electroporated with a MO mix containing 18 µg 2xChoRE reporter construct, 6 µg transfection control (pRL-CMV vector), 6 µg carrier DNA (pBluescript SK), 10 µM MO (mondoa-mis or mondoa-mo), in 50 µl dH2O at 0.29 kV, 0 W, 960 mF using a GenePulser II apparatus. Immediately after transfection cells were transferred into 4 ml PAC2 maintenance medium and resuspended. 150 µl of the cell suspension were transferred into 96 well plates containing 100 µl of maintenance medium. Cells were kept overnight at 28°C to attach. The next day cells were washed and then starved for 24 hr in starvation medium supplemented with 0.5 mM luciferin. The following day cells were treated for 24 hr with low (0.3 mM) or high glucose (12 mM) concentrations.

### Bioluminescence measurements

*In vivo* luciferase measurements of transiently transfected cells was performed at 28°C with an EnVision multilabel plate reader (PerkinElmer) as previously described (*Weger et al., 2011*; *Weger et al., 2012*), using L-15 medium without phenol red and supplemented with 0.5 mM luciferin. For HepG2 cells, DMEM medium instead of L-15 medium was used to prepare an analogous luciferin-containing medium. For *in vitro* luciferase measurements (overexpression and MO knockdown experiments), we employed the Dual-Luciferase Reporter Assay System (Promega, #E1910) as suggested by the manufacturer.

### 2-Deoxy-D-glucose (2-DG) treatment of embryos

Embryos were treated with the glucose analog 2-deoxy-glucose (2-DG, Sigma) to activate the Mondo pathway. As Glucose-6-phosphate (G6P) is thought to be the signal activating the Mondo pathway, and because 2-DG can only be metabolized to 2-DG-6-phosphate, but not further, this treatment should avoid activation of other pathways relying on further metabolization of glucose (*Li et al., 2010b*; *Stoltzman et al., 2008*).

For recording of *in vivo* bioluminescence from a ChoRE reporter, a mixture composed of the 2xChoRE luciferase reporter construct (100 ng/µl; see above, 'cell culture'), phenol red (0.1%) and either 150 mM 2-DG or water was injected into zygotes. The injected embryos were transferred into 96-well plates containing E3 medium with 0.5 mM luciferin (E3L) at high/oblong stage for bioluminescence measurement as described (*Weger et al., 2013a*; *Weger et al., 2012*; *Weger et al.,*

*2013b*). Bioluminescence was measured at sphere stage and subtracted against background luminescence from wells without embryos.

To determine whether *nsdhl* is a downstream target of MondoA and is regulated by glucose, 150 ng/µl each of *mondoa* and *mlx* transcript were injected either with water as a control or with 150 mM 2-DG. RNA was extracted at sphere stage and processed for RT-qPCR against *nsdhl*. To determine if glucose induction of *txnipa* expression was affected by *mondoa* knockdown, embryos were injected with 150 mM 2-DG or with water along with either mondoa-mis or mondoa-mo.

## CRISPR/Cas9 mediated zebrafish gene editing

CRISPR/Cas9 mediated gene editing was performed essentially as described (*Gagnon et al., 2014*), with the following modifications. Guide RNA target regions in exon 3 of *mlxip/mondoa* (5'-gagtccatattgccatccaa-3') were determined using the CHOPCHOP web tool (http://chopchop.cbu.uib.no/) (*Labun et al., 2019*). The guide RNA templates that include an SP6 promotor and Cas9 scaffold were prepared by GeneArt Strings DNA fragment gene synthesis (ThermoFisher) (5'-tataagcttccatggatt-taggtgacactatagagtccatattgccatccaagtttttagagctagaaatagcaagttaaaataaggctagtccgttatcaactt-gaaaaagtggcaccgagtcggtgcttttctagacgatgcccttgagagccttcaacccagtcctatagtgagtcgtattaggatcctac-3'). The templates were amplified by PCR and the amplification product was transcribed using the MEGAscript SP6 Transcription Kit (ThermoFisher). 60 ng/µl guide RNA were injected into 1 cell stage wild-type zebrafish embryos together with 0.9 µg/µl Cas9 protein (GeneArt Platinum Cas9 Nuclease; ThermoFisher), 0.2 M KCl and 0.1% phenol red (SIGMA-Aldrich). Guide RNA efficiency was determined using High Resolution Melting (HRM) analysis (fw: 5'-tgcctttctttcctgaagatgt-3'; rv: 5'-gctttttccattcaaaac-cagt-3'). Injected (F0) zebrafish were outcrossed with wild-type zebrafish. F1 mutant carriers were identified by HRM and PCR targeting the mutated region (PCR: fw: 5'-cgcatactcctttatcttgc-3'; rv: 5'-catgcttttttccattcaaaacc-3') on genomic DNA from fin biopsies followed by sequence analysis. Further matings of the F1 fish generated homozygous F2 fish, which were raised to adulthood. Maternal zygotic (MZ) mutants were derived from incrosses of homozygous F2 fish.

## Morpholino oligonucleotide knockdown in zebrafish embryos

Morpholino oligonucleotides (MOs) were obtained from GeneTools Inc (http://www.gene-tools.com/). MO sequences were: chrebp-mo (translation blocking MO): 5'-ttctgaatactctgcttttgccatc-3'; mondoa-mo (translation blocking MO): 5'-ggtattgtcgagtagccatgttaaa-3'; mondoa-mis (5 bp mismatch MO): 5'-ggtaatctccagtacccatcttaaa-3'; mlx-mo (translation blocking MO): 5'-cgcgctgttttccgtcattttggaa-3'; nsdhl-mo (splice blocking MO): 5'- ctgaaacaattcacacctgtttgtc-3'; p53-mo (*Robu et al., 2007*) (translation blocking MO): 5'-gcgccattgctttgcaagaattg-3'; tagged-mo (3' lissamine tagged MO of unrelated sequence): 5'-gtagcttgtactcgcattcctatct-3'. To target the transcript of interest specifically in the YSL, embryos were injected with MO into the yolk at the 1 k cell stage, as previously described (*Sakaguchi et al., 2001*). In order to label the area where the MO is localized, either lis-mondoa-mo or lis-mondoa-mis was used, or 0.08 mM of tagged-mo was added to the injection mixture. Injection of tagged-mo alone did not cause an aberrant phenotype.

Knockdown efficiency of the mondoa-mo was tested with 30 ng/µl of the *mondoa* target sequence-GFP fusion transcript encoded by the pCS2:GFP:MondoA vector. For mRNA transcription, this construct was linearized with NotI and transcribed with the mMESSAGE mMACHINE kit as recommended by the manufacturer (Ambion). Rescue experiments with *mondoa* mRNA were performed using 250 ng/µl of transcript. *mondoa* mRNA was transcribed using the mMESSAGE mMACHINE Kit (Ambion) (T7) on templates linearized with DraIII. Pregnenolone (20 µM) rescue experiments were performed as previously reported (*Hsu et al., 2006*). Knockdown efficiency of the chrebp-mo was tested with 100 ng/µl pCS2:GFP:ChREBP. Resistance of MZ*mondoa* mutant embryos to *mondoa* knockdown was tested with 0.5 mM lis-mondoa-mo and 0.5 mM lis-mondoa-mis injections into MZ*mondoa* mutants and wild-type controls.

## Confocal microscopy

For live imaging of embryos by confocal microscopy, embryos were dechorionated and transferred with a glass pipette into a chamber containing 0.1% (w/v) low melting agarose in E3. Images were taken on a Leica TCS SP5 upright microscope using either a HCX PL APO 20x/0.70 lambda blue IMM CORR or a HCX APO L 20x/0.50 W U-V-I objective at 26°C. Spatial and sensitivity resolution

were strictly kept constant between samples of comparative analysis. Reflection images using 458/488/496 nm incident light were used to assess whether a nucleus is in the EVL or YSL. The reflected light images require no labeling and provide a confocal contrast at the boundary of different tissues (*Jester et al., 1991*). We used the GFP fluorescence channel to identify the nucleus and the label-free reflection channel to visualize the extracellular matrix. Whether a nucleus belongs to EVL or YSL was individually determined in the XZ or YZ plane using the reflection channel. Images were analyzed and processed in Fiji/ImageJ (*Schindelin et al., 2012*).

To visualize endocytotic vesicles, embryos were dechorionated at five hpf and placed in 1.5% (w/v) Lucifer Yellow CH (Sigma-Aldrich, L0259) dissolved in E3 medium for 15 min (*Solnica-Krezel and Driever, 1994*). After the incubation, embryos were washed briefly with excess E3 medium and mounted in 0.5% low-melting temperature agarose for confocal imaging.

## Digital scanned laser light sheet microscopy (DSLM)

Time-lapse DSLM images of developing zebrafish embryos were acquired on a home-built microscope described in *Kobitski et al., 2015*. In the DSLM (*Keller et al., 2008*; *Schmid et al., 2013*), the sample is irradiated with two opposing beams of 488 nm laser light (LuxX 488–60 laser, Omicron-Laserage Laserprodukte GmbH, Rodgau, Germany) that were scanned up and down so as to excite fluorescence in the samples in a slab ('light sheet') with an average thickness of 6.6 µm across the field of view (1038 × 875 µm$^2$). Fluorescence emission from the illuminated region was collected perpendicular to the light sheet by using a water dipping objective (16x/0.8 w, Nikon) and projected on a 5.5 megapixel sCMOS camera (Neo, Andor, Belfast, UK), yielding 1.5 µm lateral image resolution. Image stacks of 500 frames at 2 µm displacement were collected at a rate of 20 frames per second by moving the sample through the light sheet. To achieve optimal image quality across the entire sample, images from two opposing views were acquired for each light sheet position by rotating the samples by 180˚. Consequently, a complete 3D image stack was collected every 60 s.

For DSLM imaging, dechorionated embryos were embedded in 0.1% (w/v) low melting agarose in 1x E3 medium, mounted in fluorinated ethylene propylene tubes as previously described (*Kaufmann et al., 2012*), and placed inside a sample chamber. During image acquisition over 15 hr, the temperature in the sample chamber was kept constant at 26˚C.

## Automated image processing and analysis

Fluorescently labeled nuclei were automatically extracted from the 3D image stacks using the fast segmentation approach described previously (*Stegmaier et al., 2014*). In a post-processing step, the extracted features from complementary rotation images were registered using a bead-based registration technique (*Preibisch et al., 2010*). Moreover, redundant nuclei were fused on the basis of spatial overlap of segments; a feature based object rejection was performed to minimize the amount of false positive detections. Nucleus counts reflect the number of remaining objects after information fusion and object rejection.

To track the detected nuclei over multiple time points, we made use of a nearest neighbor tracking algorithm implemented in the open-source MATLAB toolbox SciXMiner (*Mikut et al., 2017*; *Stegmaier et al., 2012*). To attain more comprehensive tracks, we used an additional heuristic to reconstruct nuclei that were missing in a single frame and we fused extracted sub-tracks in the spatio-temporal domain by applying an additional nearest neighbor matching on the start and end points of sub-tracks. It should be noted, though, that these tracks did not necessarily reflect perfect tracks on the single cell level over the whole duration, but merely are used to identify and trace the qualitative movements of nuclei in certain regions.

To compute the cell density distribution, we counted the number of nuclei located within a bounding sphere with a radius of 40 µm centered on the nucleus of interest. For visualization, we aligned the virtual embryo by manually centering it on the origin and rotating the vegetal pole such that it was centered on the positive y-axis and finally converting the Cartesian coordinates into a spherical projection using azimuth, elevation and radius (see schematic in *Figure 3B*). This representation allows us to unwrap the embryo by plotting the azimuth (range: [-pi, pi]) and the elevation (range: [-pi/2, pi/2]) in a 2D graph.

For speed determination of ingressing deep cells, we manually selected a minimum number of ten cells for each condition and tracked their movement over a time window of 100 frames, that is,

for 200 min. We included only internalizing deep cells which could be tracked over the entire time frame in both conditions. The time point of ingression onset was visually determined using slice-based maximum intensity projection videos. The estimated speed reflects the arithmetic mean of approximately 1000 displacement steps of the selected cells during the time interval.

### Implementation details

Maximum intensity projections, seed detection and segmentation were performed using a custom-built C++ application based on the Insight Toolkit (ITK). To speed up calculations, we processed multiple image stacks in parallel on a Hadoop cluster containing 25 nodes (each equipped with two quad-core Intel Xeon E5520 CPUs @ 2.27 GHz and 36 GB of memory) of the Large Scale Data Facility (LSDF) of the KIT (*Garcia et al., 2011*). Tracking, data analysis and visualizations were performed using MATLAB with the open-source toolbox SciXMiner (*Mikut et al., 2017*; *Stegmaier et al., 2012*) and Fiji (*Schindelin et al., 2012*).

## Metabolite quantification

ATP, ADP and AMP levels in mondoa-mo and mondoa-mis injected embryos were determined at sphere stage, just before the morphological phenotype becomes apparent, using a mass spectrometry approach. Metabolites were extracted from three pools of 30 zebrafish embryos of each condition with 0.3 ml of 0.1 M HCl in an ultrasonic ice bath for 10 min. The resulting homogenates were centrifuged twice for 10 min at 4°C and 16.400 g to remove cell debris. AMP, ADP and ATP were derivatized with chloroacetaldehyde as described (*Bürstenbinder et al., 2007*) and separated by reversed phase chromatography on an Acquity BEH C18 column (150 mm x 2.1 mm, 1.7 µm, Waters) connected to an Acquity H-class UPLC system. Prior to separation, the column was heated to 42°C and equilibrated with five column volumes of buffer A (5.7 mM TBAS, 30.5 mM KH2PO4 pH 5.8) at a flow rate of 0.45 ml min$^{-1}$. Separation of adenosine derivates was achieved by increasing the concentration of buffer B (2/3 acetonitrile in 1/3 buffer A, by volume) in buffer A as follows: 1 min 1% B, 1.6 min 2% B, 3 min 4.5% B, 3.7 min 11% B, 10 min 50% B, and return to 1% B in 2 min. The separated derivatives were detected by fluorescence (Acquity FLR detector, Waters, excitation: 280 nm, emission: 410 nm, gain: 100) and quantified using ultrapure standards (Sigma). Data were acquired and processed with the Empower3 software suite (Waters). Energy charge values were calculated for each batch of embryos separately according to ([ATP] + 0.5 [ADP])/([ATP] + [ADP] + [AMP]) and were analyzed for differences between conditions by One-Way ANOVA.

## Next-generation sequencing

### Generation and sequencing of mRNA and total RNA libraries

RNA-seq analysis was performed on embryos injected with mondoa-mo or mondoa-mis as well as on MZ*mondoa* mutants affected or unaffected in epiboly and wild-type embryos. MO injected embryos were sampled when control embryos were at sphere stage (age-matched), mutant and wild-type embryos when affected mutants could be clearly distinguished (90% epiboly of controls). Triplicate total RNA samples of each condition were used to generate cDNA libraries. The quality and integrity of the sampled total RNA were assessed with a Bioanalyzer 2100 (Agilent). Sample concentrations were measured using a NanoDrop spectrometer (ThermoScientific). Sequencing libraries were prepared from total RNA with a RNA integrity number (RIN) greater than 8.0. Indexed mRNAseq libraries were generated from morphant samples following the TruSeq mRNA kit protocol (Illumina). Libraries for mutant samples were prepared using the TruSeq Stranded Total RNA Sample Prep Kit with Ribo-Zero Gold depletion (Illumina). Libraries of the morphant experiments were sequenced on a HiSeq 1000 system (Illumina) in 3 lanes of a paired-end flow cell. For each end, 50 nucleotides were sequenced. Cluster and sequencing reactions were performed with the TruSeq SBS kit v3-HS and the paired-end cluster v3-cBot-HS kit. The libraries of the MZ*mondoa* mutants were sequenced on Illumina Hiseq 4000 sequencer as Paired-End 50 base reads following Illumina's instructions. Image analysis and base calling were performed using RTA 2.7.7 and bcl2fastq2.17.1.14.

### Mapping and differential gene expression analysis

Paired-end reads were mapped onto the *Danio rerio* genome assembly (GRCz11/danRer11) using STAR 2.3.8 (*Dobin et al., 2013*). Uniquely mapped reads were counted for each gene locus defined

in the Ensembl 96 release. Briefly, we applied SAMtools view (*Li et al., 2009*) to assign uniquely mapped paired-end reads. For downstream applications such as data visualization, we normalized the count data by size factor and applied a variance stabilizing transformation (*Huber et al., 2002*). Moderated log fold changes were calculated as proposed by *Love et al., 2014*. DESEq2 (*Love et al., 2014*) was used to assess differential gene expression between two conditions. To compare the gene expression signatures of MZ*mondoa* mutants with and without epiboly phenotype to those of *mondoa* morphants and their respective controls, we employed the rank-rank hypergeometric overlap (RRHO) algorithm (*Plaisier et al., 2010*). Competitive gene set enrichment analyses were performed using *CAMERA* (*Wu and Smyth, 2012*). GO term annotations were included in the analysis. In addition, a manually curated set of zebrafish genes linked with important metabolic pathways assembled from KEGG and Ensembl annotations (*Weger et al., 2016*) complemented the approach. We visualized gene set enrichment using the barcodeplot function from the limma package (*Ritchie et al., 2015*). Raw files and technical details about the RNA-Seq data have been deposited in NCBI's Gene Expression Omnibus (*Edgar et al., 2002*) and are accessible through GEO Series accession number GSE144350 (https://www.ncbi.nlm.nih.gov/geo/query/acc.cgi?acc=GSE144350).

## Statistical analysis

Bioluminescence measurements in embryos were analyzed using Mann–Whitney U test, as the data were not normally distributed. Data sets with a normal distribution, including RT-qPCR data, velocity comparison of ingressing deep cells, epiboly progression, cell based bioluminescence measurements and energy charge of embryos, were analyzed with a Student's t-test (for two groups) or ANOVA (for more than two groups). Multiple testing was corrected by the Benjamini-Hochberg procedure (*Benjamini and Hochberg, 1995*) or an adequate ANOVA posttest. Bonferroni's posttest was chosen if all groups had to be compared, while Dunnett's posttest was chosen if the groups were compared only to the control. Data were analyzed and visualized by GraphPad software (GraphPad Software, Inc), Microsoft Excel or R (https://www.r-project.org/). 'n' indicates number of biological repeats. Asterisks indicate *p*-values of *$p \leq 0.05$, **$p \leq 0.01$ and ***$p \leq 0.001$.

## Acknowledgements

We thank Ferenc Müller for critically reading the manuscript and Sepand Rastegar, Nicholas S Foulkes, Marco Ferg and Gerald Brenner-Weiß for helpful discussions. Sepand Rastegar and Marcel Schaaf kindly provided plasmids. We acknowledge Christin Lederer and Giuseppina Pace for excellent technical assistance. Olivier Armant, Victor Gourain and colleagues of the NGS core facility of the BIFTM program and Bernard Jost and colleagues of the GenomEast platform at the IGBMC performed library preparations and sequencing of the in the mRNA-seq study and total RNA-seq study, respectively. Gernot Poschet of the Metabolomics Core Technology Platform of the Excellence cluster 'CellNetworks' (University of Heidelberg) ran the HPLC-based metabolite quantification, supported by the Deutsche Forschungsgemeinschaft (grant ZUK 40/2010–3009262). Ariel Garcia and Serguei Bourov (Steinbuch Centre for Computing at KIT, Helmholtz Program Supercomputing) as well as Volker Hartmann (Institute for Data Processing and Electronics, KIT) provided support for data handling and Hadoop processing. Funding was provided by the Studienstiftung des deutschen Volkes (to MW), by a Marie Curie Intra-European fellowship (grant PIEF-GA-2013–625827, to MW), by the Excellence Cluster 'CellNetworks' (University of Heidelberg, to TD), by the Effect-Net research network in water research of the Ministerium für Wissenschaft, Forschung und Kunst Baden-Württemberg (to TD), by the Deutsche Forschungsgemeinschaft (DFG grant DI913/4-1, to TD), by the Association française contre les myopathies (AFM, to TD), by EU FP7-ZFHEALTH-2007-B2 (to US), by the Interreg Network for Synthetic Biology in the Upper Rhine valley (NSB-Upper Rhine), and by the Helmholtz Programs BioInterfaces in Technology and Medicine (BIFTM) and Science and Technology of Nanostructures (STN).

# Additional information

## Competing interests

Benjamin D Weger: BDW was an employee of Nestlé Health Sciences SA. Cédric Gobet: CG was an employee of Nestlé Health Sciences SA. Alice Parisi: AP is an employee of Nestlé Health Sciences SA. Frédéric Gachon: FG was an employee of Nestlé Health Sciences SA. Philipp Gut: PG is an employee of Nestlé Health Sciences SA. The other authors declare that no competing interests exist.

## Funding

| Funder | Grant reference number | Author |
|---|---|---|
| Studienstiftung des Deutschen Volkes | | Meltem Weger |
| Marie Curie Intra-European Fellowship | PIEF-GA-2013-625827 | Meltem Weger |
| Excellence cluster "CellNetworks" | | Thomas Dickmeis |
| Effect-Net research network in water research, Ministerium für Wissenschaft, Forschung und Kunst Baden-Württemberg | | Thomas Dickmeis |
| Association Française contre les Myopathies | | Thomas Dickmeis |
| EU FP7-ZFHEALTH-2007-B2 | | Uwe Strähle |
| Interreg Network for Synthetic Biology in the Upper Rhine Valley | | Uwe Strähle |
| Helmholtz-Gemeinschaft | BioInterfaces in Technology and Medicine (BIFTM) | Uwe Strähle Gerd Ulrich Nienhaus Ralf Mikut Thomas Dickmeis |
| Helmholtz-Gemeinschaft | Science and Technology of Nanostructures (STN) | Gerd Ulrich Nienhaus |

The funders had no role in study design, data collection and interpretation, or the decision to submit the work for publication.

## Author contributions

Meltem Weger, Conceptualization, Resources, Data curation, Formal analysis, Supervision, Funding acquisition, Validation, Investigation, Visualization, Methodology, Writing - original draft, Writing - review and editing; MW performed most of the experiments corresponding to Figures 1-6 and their supplements (i.e., characterization of the Mondo pathway in zebrafish and mondoa knockdown studies), analyzed these experiments, performed DSLM imaging and/or analysis, analysed the experiments underlying Figure 1-figure supplement 1G, Figure 7, Figure 7- figure supplement 1 and Figure 7- figure supplement 2, and performed statistical analyses; Benjamin D Weger, Conceptualization, Resources, Data curation, Software, Formal analysis, Supervision, Validation, Investigation, Visualization, Methodology, Writing – original draft, Writing - review and editing; BDW performed most of the experiments corresponding to Figures 1-6 and their supplements (i.e., characterization of the Mondo pathway in zebrafish and mondoa knockdown studies), analyzed these experiments, performed DSLM imaging and/or analysis, generated the mondoa mutant, analysed the experiments underlying Figure 1-figure supplement 1G, Figure 7, Figure 7- figure supplement 1 and Figure 7- figure supplement 2, analysed RNA-seq studies, and performed statistical analyses; Andrea Schink, Formal analysis, Validation, Investigation, Visualization, Writing - review and editing; AS generated the mondoa mutant, performed the experiments underlying Figure 1-figure supplement 1G, Figure 7, Figure 7- figure supplement 1 and Figure 7- figure supplement 2 and analysed these experiments; Masanari Takamiya, Resources, Validation, Investigation, Visualization, Methodology, Writing -

review and editing; MT performed and analyzed the experiment in Figure 4-figure supplement 1J-O and performed DSLM imaging and/or analysis; Johannes Stegmaier, Resources, Data curation, Software, Formal analysis, Visualization, Writing - review and editing; JS performed DSLM analysis; Cédric Gobet, Resources, Data curation, Software, Formal analysis; CG analysed RNA-seq studies; Alice Parisi, Resources, Investigation; AP generated the mondoa mutant; Andrei Yu Kobitski, Resources, Methodology; AK performed DSLM imaging and/or analysis; Jonas Mertes, Investigation, Methodology; JM performed DSLM imaging and/or analysis; Nils Krone, Frédéric Gachon, Philipp Gut, Resources, Supervision, Funding acquisition, Writing - review and editing; Uwe Strähle, Resources, Supervision, Funding acquisition; Gerd Ulrich Nienhaus, Resources, Supervision, Funding acquisition, Methodology; Ralf Mikut, Resources, Software, Formal analysis, Supervision, Funding acquisition; RM performed DSLM analysis; Thomas Dickmeis, Conceptualization, Resources, Formal analysis, Supervision, Funding acquisition, Investigation, Methodology, Writing - original draft, Project administration, Writing - review and editing; TD analyzed the experiments corresponding to Figures 1-6 and their supplements, performed the experiments underlying Figure 1-figure supplement 1G, Figure 7, Figure 7- figure supplement 1 and Figure 7- figure supplement 2, and analysed these experiments. All authors discussed the results and approved the final manuscript.

### Author ORCIDs
Meltem Weger  https://orcid.org/0000-0002-9452-2373
Benjamin D Weger  https://orcid.org/0000-0002-1831-3561
Johannes Stegmaier  https://orcid.org/0000-0003-4072-3759
Gerd Ulrich Nienhaus  https://orcid.org/0000-0002-5027-3192
Frédéric Gachon  https://orcid.org/0000-0002-9279-9707
Thomas Dickmeis  https://orcid.org/0000-0002-9819-1962

### Ethics
Animal experimentation: All zebrafish husbandry was performed in accordance with the German animal protection standards and approved by the Government of Baden-Württemberg, Regierungspräsidium Karlsruhe, Germany (Aktenzeichen35-9185.64/BH KIT), by the Home Office, United Kingdom (Scientific Procedures, Act 1986), and by the Service de la consommation et des affaires vétérinaires du canton de Vaud, Switzerland.

### Decision letter and Author response
Decision letter https://doi.org/10.7554/eLife.57068.sa1
Author response https://doi.org/10.7554/eLife.57068.sa2

## Additional files

### Supplementary files
• Supplementary file 1. Differentially expressed genes identified in the RNA-seq analysis. Differentially expressed genes of mondoa-mo vs. mondoa-mis injected embryos, of MZ*mondoa* mutants with or without epiboly phenotype vs. WT embryos and of MZ*mondoa* mutants with epiboly phenotype vs. MZ*mondoa* mutants without epiboly phenotype.

• Supplementary file 2. Metabolic pathway and GO term enrichment analysis for differentially expressed genes identified in the RNA-seq analysis.

• Transparent reporting form

### Data availability
Raw files and technical details about the RNA-Seq data have been deposited in NCBI's Gene Expression Omnibus and are accessible through GEO Series accession number GSE144350 (https://www.ncbi.nlm.nih.gov/geo/query/acc.cgi?acc=GSE144350).

The following dataset was generated:

| Author(s) | Year | Dataset title | Dataset URL | Database and Identifier |
|---|---|---|---|---|
| Weger M, Weger BD, Schink A, Takamiya M, Stegmaier J, Gobet Cd, Parisi A, Kobitski AY, Mertes J, Krone N, Strähle U, Nienhaus GU, Mikut R, Gachon F, Gut P, Dickmeis T | 2020 | Regulation of cholesterol biogenesis by the glucose-sensing transcription factor MondoA is required for zebrafish epiboly | https://www.ncbi.nlm.nih.gov/geo/query/acc.cgi?acc=GSE144350 | NCBI Gene Expression Omnibus, GSE144350 |

The following previously published dataset was used:

| Author(s) | Year | Dataset title | Dataset URL | Database and Identifier |
|---|---|---|---|---|
| White RJ, Collins JE, Sealy IM, Wali N, Dooley CM, Digby Z, Stemple DL, Murphy DN, Billis K, Hourlier T, Füllgrabe A, Davis MP, Enright AJ, Busch-Nentwich EM | 2017 | A high-resolution mRNA expression time course of embryonic development in zebrafish | http://doi: 10.7554/eLife.30860 | ENA, accession-number-ERP014517 |

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
