## [Decision Letter]

**Acceptance summary:**

The study by Weger and coworkers explores the role of conserved intracellular sugar sensor MondoA-Mlx in zebrafish early development. The authors demonstrates for the first time that MondoA regulates the cholesterol biosynthesis pathway by extensively analyzing morphants and mutants in zebrafish.

**Decision letter after peer review:**

Thank you for submitting your article "Regulation of cholesterol biosynthesis by glucose-sensing MondoA is required for zebrafish epiboly" for consideration by *eLife*. Your article has been reviewed by four peer reviewers, including Koichi Kawakami as the Reviewing Editor and Reviewer #4, and the evaluation has been overseen by Marianne Bronner as the Senior Editor. The following individuals involved in review of your submission have agreed to reveal their identity: Masahiko Hibi (Reviewer #1); Donald E Ayer (Reviewer #2).

The reviewers have discussed the reviews with one another and the Reviewing Editor has drafted this decision to help you prepare a revised submission.

Summary:

As you can see reviewers' comments. they think your findings, the function of MondoA in early vertebrate embryogenesis and the discovery of the novel MondoA-nsdhl (pregnenolone) pathway are important. However they also pointed out the weakness of the study, namely the difference in the gene regulation (terpenoid biosynthesis genes) in mutants and morphants, and insufficient analysis of the compensation mechanism.

Essential revisions:

The reviewers think it is important to show (1) the epiboly phenotype observed in some of the MZ mutant embryos is indeed caused by the same mechanism as shown in the morphants, namely by down-regulation of the cholesterol synthesis genes and (2) upregulation of the terpenoid pathway is responsible for the compensation (rescue) of the epiboly phenotype seed in the morphants and mutants. In the revised version, the reviewers requested that these issues should be addressed. If you cannot address these by more extensive informatics analysis of the existing RNA-seq data, additional experiments may be needed.

*Reviewer #1:*

The manuscript by Weger et al. describes a role of glucose-sensing MondoA in cholesterol/pregenenolon biogenesis-dependent control of epiboly movement in zebrafish. The authors demonstrate that antisense morpholino (MO)-mediated knockdown of *mondoa* led to defective epiboly. Furthermore, MO-mediated knockdown of nsdhl, which encodes an enzyme of cholesterol and pregnenolone synthesis and functions downstream of MondoA, also resulted in similar epiboly defects. Pregnenolon was previously reported to be involved in formation of microtubules and microtubule-dependent epiboly. The authors show treatment with pregnenolone rescues defective epiboly in in *mondoa* and *nsdhl* morphants. They conclude that the ModoA-Nsdhl pathway controls epiboly through upregulation of pregnenolon. The manuscript could be the first report showing involvement of the glucose-sensing MondoA system in early vertebrate development. They show a lot of data that support their conclusion. Quality of the data is high. The manuscript is well written.

My major concern is partial rescue of *mondoa* morphants by wild-type *mondoa* mRNA (Figure 2E and subsection “Loss of *mondoa* function leads to severe epiboly defects”) and of mondoa/*nsdhl* morphants by the treatment with pregnenolone (Figure 5, subsection “Pregnenolone partially rescues epiboly in *mondoa* morphants and normalizes microtubule patterns”). These data raise the question whether effects of MO-mediated knockdown are really specific. However, the authors demonstrate that injection of MO did not induce defective epiboly in *MZmondoa* mutants, although only a small number of *MZmondoa* mutants showed epiboly defects. This experiment satisfies the recent criteria for MO experiments. Analyses of maternal and maternal zygotic mutants are not easy. The authors tried to explain the low penetrance by gene expression profiling. Although the authors did not show data from *nsdhl* mutants, I believe that the authors have shown compelling evidence for their conclusion. I would not ask them to do additional experiments.

*Reviewer #2:*

This is an interesting and well-done paper that presents a characterization of the glucose sensing transcription factors, MondoA, Mlx and ChREBP in zebrafish epiboly. This is the first characterization of this transcription family in zebrafish, so it will be of interest to that community of developmental biologists. It is uncertain whether it will have broad appeal to the general audience of *eLife*. The analysis shows that MondoA and Mlx function at the mechanistic level much like their mammalian counterparts as glucose responsive transcription factors. A series of morpholino experiments connect MondoA to epiboly via the regulation of *nsdhl* and ultimately steroid biosynthesis. The authors then develop and partially characterize a genetic mutant of MondoA, which doesn't phenocopy the morphant phenotype in epiboly, or only does so with low penetrance. Their claim that a compensatory mechanism involving upregulation of genes involved in terpenoid biosynthesis. They demonstrate that several of the pathway genes are upregulated in the MZMondoA mutant, validating two, but there is no functional characterization of the putative compensatory mechanism. The discussion is balanced and the authors argue that similar mechanisms could be at play in mammals. It is true that phenotypes of MondoA, and ChREBP have only been characterized in adults (there is paper looking at Mlx in skeletal muscle that should be referenced). Whole body knockouts of ChREBP (published) and MondoA/Mlx(unpublished) don't display developmental defects, so the mechanisms described in the zebrafish system may not extend to mammalian systems.

It would be interesting to know whether the regulation of *nsdhl* and other cholesterol biosynthetic genes is direct and contain ChoREs. This could be accomplished with a more thorough bioinformatic analysis. It would also be nice to know if the genes induced by 2-DG treatment are also MondoA dependent.

*Reviewer #3:*

The study of Weger and coworkers explores the role of conserved intracellular sugar sensor MondoA-Mlx in zebrafish early development. The study shows that transient knockdown of MondoA and its necessary binding partner Mlx lead to impaired morphogenetic movement of epiboly in the embryos. The authors also use (likely null) mutants of MondoA and observe similar defects, however with very low penetrance. Rnaseq analysis of MondoA knockdown embryos revealed strong downregulation of nsdhl, encoding an enzyme involved in cholesterol biosynthesis. Knockdown of *nsdhl* showed similar epiboly phenotype as MondoA knockdown, implying that defect of cholesterol biosynthesis is the underlying reason for the developmental defect. This is supported by the finding that treatment with steroid hormone pregnenolone partially rescues the MondoA phenotype.

This study is of high interest to the field, and therefore possibly suitable for *eLife*. It is to my knowledge the first characterization of MondoA-Mlx in vivo function in fish, it provides unprecedented evidence for MondoA-Mlx in early embryogenesis, and it establishes MondoA-Mlx as a putative regulator of steroid hormone metabolism. In general, the experiments have been carefully designed, conducted and documented and the conclusions are sound.

I have a couple of requests to strengthen the central findings of the manuscript that should be addressed prior to publication.

1) While *nsdhl* knockdown produces similar phenotype to MondoA knockdown, it remains unclear whether the epiboly phenotype can be mainly explained by deregulation of this one target gene. To test this, the authors should test whether transgenic *nsdhl* is sufficient to rescue the MondoA knockdown phenotype.

2) The partial rescue of MondoA knockdown by pregnenolone supports the idea that the phenotype is due to defects in steroid biosynthesis. The authors should directly analyze the levels of cholesterol (for example cholesterol binding fluorescent dyes) to test whether the biosynthesis is indeed affected by MondoA knockdown.

3) RNAseq data should be more thoroughly analyzed and represented. Gene set enrichment analyses should be done to show, which pathways and processes are affected by MondoA knockdown. It would be also interesting to know, whether the conserved MondoA targets (shown in Figure 1) were affected by MondoA knockdown or MondoA mutant.

4) The differences between MondoA knockdown and MondoA mutant are the most puzzling part of this manuscript. I have several requests regarding this part:

– The authors performed an RNAseq of mutant embryos without an epiboly phenotype and state that "Surprisingly, *nsdhl* mRNA levels were not altered" (data not shown). It remains unclear to me whether this means not altered from MondoA knockdown (downregulated) or not altered from control (not downregulated). Please clarify this (and show the data). If not downregulated compared to control, it would be important to see what is the expression of *nsdhl* mRNA in mutants that show epiboly phenotype.

– The authors also show that a set of genes involved in terpenoid biosynthesis is differentially expressed in mutants (upregulated) and knockdown (downregulated) and suggest this would be the reason for the lacking phenotype. How are these genes expressed in MondoA mutant embryos with an epiboly phenotype? Does their knockdown cause an epiboly phenotype?

– In order to validate the effects of the genetic perturbations, a systematic comparison between the gene expression changes in the MondoA knockdown vs. mutants should be represented. Are other gene groups similarly deregulated in the two conditions and only the terpenoid group shows differential expression?

*Reviewer #4:*

In this manuscript, the authors analyzed the function of MondoA in vivo in zebrafish. They found that knockdown of MondoA caused the epiboly defects. Then they analyzed the morphant phenotypes extensively, and found defects in the YSL structure or function. Next, they performed RNA -seq analysis, and identified the *nsdhl* gene as a target, which is important for cholesterol and pregnenolone synthesis. The Nsdhl morphants showed a similar phenotype as the MondoA morpants, and the MondoA morphant phenotypes were partially rescued by pregnenolone, thus revealing a novel MondoA-pregnenolone pathwaty. Furthermore, they showed that during epiboly pregnenolone controls stabilization of yolk microtubules. Finally, they created knockout mutant of MondoA by CRISPR/Cas9. Homozygotes are viable and they created maternal zygotic MondoA mutant embryos. They saw the epiboly phenotype in smaller percentages of MZ embryos. Unlike the morphants, in MZ embryos the terpenoid pathway genes are upregulated. This is the first demonstration of how MondoA knockdown affects embryonesis in details in a vertebrate.

Major concerns:

1) Rescue efficiencies by MondoA mRNA injected to MondoA morphants seem low.

2) They tried to interprete the difference between MondoA morphants and knockout mutants by an unknown compensatory mechanism. I think the argument was not strong enough. The authors at least need to show that upregulation of the terpenoid pathway can rescue the epiboly phenotype seen in morphants.

[Editors' note: further revisions were suggested prior to acceptance, as described below.]

Thank you for submitting your article "Regulation of cholesterol biosynthesis by glucose-sensing MondoA is required for zebrafish epiboly" for consideration by *eLife*. Your article has been reviewed by four peer reviewers, including Koichi Kawakami as the Reviewing Editor and Reviewer #4, and the evaluation has been overseen by Marianne Bronner as the Senior Editor. The following individuals involved in review of your submission have agreed to reveal their identity: Masahiko Hibi (Reviewer #1); Donald E Ayer (Reviewer #2).

The reviewers have discussed the reviews with one another and the Reviewing Editor has drafted this decision to help you prepare a revised submission.

The manuscript has been significantly revised. In the revised version, the authors extensively performed transcriptome analysis of morphants and mutants with or without the phenotype, and identified overlap between morphants and mutants and differences between mutants with and without the phenotype. Unexpectedly expression of *nsdhl* was not downregulated in the mutants with or without the phenotype. The authors found that mutants with unaffected epiboly showed stronger upregulation of genes in the terpenoid biosynthesis pathway and steroid hormone biosynthesis, compared to mutants with affected epiboly. The data strongly suggest that the upregulation of these genes compensated for Mondoa deficiency in the mutants, supporting the conclusion that MondoA-mediated cholesterol biosynthesis plays an important role.

Major concerns:

1) A main concern with the original manuscript version was that there was no downstream mechanism for the epiboly phenotype presented, which would be consistently supported by evidence obtained from both morpholino and mutant experiments. With the new RNAseq dataset the gene expression output of the morphants and the mutants with affected epiboly can be compared. The morphants show significant downregulation of genes in the terpenoid backbone and sterol biosynthesis pathways (including the most strongly downregulated gene nsdhl), supporting the main conclusion. The mutants with affected epiboly, however, show no change in sterol biosynthesis (including nsdhl) or the terpenoid backbone biosynthesis pathways. The authors draw attention to the upregulation of the terpenoid backbone biosynthesis pathway in the unaffected mutants (in contrast to the affected mutants) and imply that these genes have a compensatory role. However, experimental evidence for such a compensatory effect is not presented. It also remains unclear to me what would these genes compensate for, since no impairment of cholesterol or steroid hormone metabolism is observed in the mutants. Accordingly, this should be discussed. Further, the main conclusion is "regulation of cholesterol biosynthesis by glucose-sensing MondoA is required for zebrafish epiboly” is not sufficiently supported by the *mondoa* mutant data presented. From this aspect, the title better be modified.

2) (similar to the #1) The fact that expression of *nsdhl* was not altered in the MZ mutants with or without the phenotype should be discussed (in Discussion). The authors should address why they do not think the transcriptome analysis of morphants were artifacts.

3) The section "*MZmondoa* mutants show compensatory changes in expression of cholesterol/steroid biosynthesis genes" was really difficult to read. I think the authors better describe the results as they described in the reply mail to us.

Also subsection “Maternal-zygotic mutants of *mondoa* show partially penetrant epiboly phenotypes”: “an epiboly phenotype” should be “the epiboly phenotype”.

Subsection “*MZmondoa* mutants show compensatory changes in expression of cholesterol/steroid biosynthesis genes”: “*MZmondoa* embryos with and without an epiboly phenotype”… This should be: *MZmondoa* embryos with and without the mutant phenotype, or *MZmondoa* embryos with severe and no mutant phenotype or *MZmondoa* embryos with severe and no epiboly anomalies.

Also in the same section: “both phenotypes of MZ mutants” should be “the MZ mutants of both phenotypes”

“… as an important component of the compensation response”. This is a compensation response specifically required to restore the phenotype.

---

## [Author Response]

Essential revisions:The reviewers think it is important to show (1) the epiboly phenotype observed in some of the MZ mutant embryos is indeed caused by the same mechanism as shown in the morphants, namely by down-regulation of the cholesterol synthesis genes and (2) upregulation of the terpenoid pathway is responsible for the compensation (rescue) of the epiboly phenotype seed in the morphants and mutants. In the revised version, the reviewers requested that these issues should be addressed. If you cannot address these by more extensive informatics analysis of the existing RNA-seq data, additional experiments may be needed.

To characterize more comprehensively transcriptome changes in the MZ mutant embryos, as suggested by the reviewers, we have now added to the revised version of the manuscript RNA-seq data of MZ*mondoa* mutants that show an epiboly phenotype. Furthermore, we added an unbiased global gene set enrichment analysis to the manuscript that highlights the importance of cholesterol related pathways for compensation of the epiboly phenotype. These results are shown in a set of new figures and/or figure panels (new Figure 7, Figure 7—figure supplement 1 and Figure 7—figure supplement 2) and a new Supplementary file 2.

To compare the gene expression signatures of MZ*mondoa* mutants with and without epiboly phenotype to those of *mondoa* morphants and their respective controls, we employed the rank-rank hypergeometric overlap (RRHO) algorithm (Plaisier et al., 2010). As shown in Figure 7—figure supplement 2A-D, the analysis reveals that differential gene expression significantly overlaps between mutants and morphants, demonstrating that both conditions are reflecting a lack of MondoA function. Specifically, mutant embryos show a significant overlap in differential gene expression with respect to the morphants mainly in the upregulated gene fraction. There are also some differences apparent between the mutants with and without an epiboly phenotype. One observes that the overlap in downregulated genes is weaker between morphants and mutants without epiboly phenotype (compare lower left quadrant in B and C). This observation indicates that mainly the downregulated genes in *mondoa* morphants are transcriptionally compensated in MZ*mondoa* mutants with unaffected epiboly. In contrast, the affected mutants show a weaker overlap in the upregulated gene signatures, indicating that only a fraction of the upregulated genes in MZ*mondoa* mutants is actually important for phenotype compensation.

We next explored differences in gene expression between the two mutant phenotypes to reveal potential mechanisms for epiboly compensation. We turned our attention to genes in the cholesterol biosynthesis pathway based on our previous observations in morphant embryos. Our analysis showed that *nsdhl* gene expression is comparable to wild-type embryos in both MZ*mondoa* mutant phenotypes (affected and unaffected epiboly) and, thus, is compensated in both (Supplementary file 1 and Figure 7—figure supplement 2E). To better illustrate the expression pattern of *nsdhl* across the different conditions, we now have added the additional graphs as Figure 7—figure supplement 2F. Therefore, differences in *nsdhl* expression between MZ*mondoa* mutants cannot explain the differences in the epiboly phenotype. This observation indicates that the restoration of expression to wild-type levels of this most downregulated gene upon *mondoA* knockdown by itself is not sufficient for compensation. Consistently, in both mutant phenotypes the entire sterol biosynthesis pathway of which *nsdhl* forms part, lacks the coordinated downregulation that was seen in the morphants (unaffected: p= 0.87, affected: p= 0.79; Figure 7F and Figure 7—figure supplement 1A).

But what is causing the epiboly arrest if MZ*mondoa* mutants with affected epiboly share such compensatory transcriptional signatures with MZ*mondoa* mutants without epiboly phenotype? *mondoa* morphants show a significant downregulation in terpenoid biosynthesis (p=0.016) as well as in sterol biosynthesis pathways (p=0.022; Top 2 downregulated pathways in a global gene set enrichment analysis, Supplementary file 2). As indicated in the previous version of the manuscript, we observed a broad compensatory upregulation of gene expression in the terpenoid backbone biosynthesis pathway upstream of cholesterol biosynthesis: MZ*mondoa* mutants with unaffected epiboly show a coordinated upregulation of terpenoid pathway genes (i.e., they show a statistically significant enrichment for upregulation in a directional gene set enrichment test; p=0.012; Figure 7F-G, Figure 7—figure supplement 1B and Supplementary file 2). Intriguingly, mutants with affected epiboly lack this significant coordinated upregulation in the terpenoid backbone biosynthesis pathway that is present in unaffected mutants (p=0.178; Figure 7F-G and Figure 7—figure supplement 1B). This key difference highlights the terpenoid backbone biosynthesis pathway as an important component of the compensation response.

As we have likewise reported in the previous version of the manuscript, gene expression of steroid hormone biosynthesis pathway genes shows a coordinated upregulation in MZ*mondoa* mutants with unaffected epiboly (p=0.033; Figure 7F and Figure 7—figure supplement 1C), probably to compensate for reduced flux through the biosynthesis pathways upstream. This upregulation is also shown by MZ*mondoa* mutants with epiboly phenotype (p=0.017; Figure 7F and Figure 7—figure supplement 1C). Strikingly, it is even increased when compared to the mutants with unaffected epiboly (p=0.049), and thereby can lead to strongly increased expression levels compared to wild-type. This is illustrated by the expression of *cyp11a1*, the enzyme that is immediately upstream of pregnenolone synthesis (Figure 7—figure supplement 2G ). However, in the absence of coordinated regulation of genes in the terpenoid pathway, this regulation is apparently not sufficient to rescue epiboly progression.

In summary, despite some successful compensatory gene expression regulations, MZ*mondoa* mutants with affected epiboly fail to properly regulate the entire set of genes necessary for epiboly progression. They are not lacking compensation *per se*, but their compensation is inadequate and fails. We have implemented these results into the revised version of the manuscript.

Reviewer #1:[…] My major concern is partial rescue of mondoa morphants by wild-type mondoa mRNA (Figure 2E and subsection “Loss of mondoa function leads to severe epiboly defects”) and of mondoa/nsdhl morphants by the treatment with pregnenolone (Figure 5, subsection “Pregnenolone partially rescues epiboly in mondoa morphants and normalizes microtubule patterns”). These data raise the question whether effects of MO-mediated knockdown are really specific. However, the authors demonstrate that injection of MO did not induce defective epiboly in MZmondoa mutants, although only a small number of MZmondoa mutants showed epiboly defects. This experiment satisfies the recent criteria for MO experiments. Analyses of maternal and maternal zygotic mutants are not easy. The authors tried to explain the low penetrance by gene expression profiling. Although the authors did not show data from nsdhl mutants, I believe that the authors have shown compelling evidence for their conclusion. I would not ask them to do additional experiments.

We thank reviewer #1 for the appreciation that we have performed our MO experiments according to “the recent criteria for morpholino experiments” and that we provide “compelling evidence” for our conclusions.

Reviewer #2:This is an interesting and well-done paper that presents a characterization of the glucose sensing transcription factors, MondoA, Mlx and ChREBP in zebrafish epiboly. This is the first characterization of this transcription family in zebrafish, so it will be of interest to that community of developmental biologists. It is uncertain whether it will have broad appeal to the general audience of eLife. The analysis shows that MondoA and Mlx function at the mechanistic level much like their mammalian counterparts as glucose responsive transcription factors. A series of morpholino experiments connect MondoA to epiboly via the regulation of nsdhl and ultimately steroid biosynthesis. The authors then develop and partially characterize a genetic mutant of MondoA, which doesn't phenocopy the morphant phenotype in epiboly, or only does so with low penetrance. Their claim that a compensatory mechanism involving upregulation of genes involved in terpenoid biosynthesis. They demonstrate that several of the pathway genes are upregulated in the MZMondoA mutant, validating two, but there is no functional characterization of the putative compensatory mechanism. The discussion is balanced and the authors argue that similar mechanisms could be at play in mammals. It is true that phenotypes of MondoA, and ChREBP have only been characterized in adults (there is paper looking at Mlx in skeletal muscle that should be referenced). Whole body knockouts of ChREBP (published) and MondoA/Mlx(unpublished) don't display developmental defects, so the mechanisms described in the zebrafish system may not extend to mammalian systems.

We have now added the reference to the *mlx* mutant mice (Hunt et al., 2015) into the introductory part of the revised version of the manuscript.

It is certainly possible that the MondoA function in early development which we report is species specific. However, given the strong genetic compensation that we observed even in our maternal-zygotic mutants, it is equally possible that such genetic compensation occurs in mammals as well. These are interesting questions for future work, both from a mechanistic and an evolutionary viewpoint.

It would be interesting to know whether the regulation of nsdhl and other cholesterol biosynthetic genes is direct and contain ChoREs. This could be accomplished with a more thorough bioinformatic analysis. It would also be nice to know if the genes induced by 2-DG treatment are also MondoA dependent.

Indeed, MondoA might regulate the expression of *nsdhl* directly or indirectly. To address this question, we have performed cell culture studies using a reporter gene construct containing a 1 kbp portion of the *nsdhl* promoter. This reporter did not show glucose induced reporter gene activity in the presence of *mondoa* and *mlx* RNA (Author response image 1). Neither could we detect putative ChoRE enhancer sequences in this 1 kbp region by *in silico* analysis using FIMO within the MEME suit (Grant, Bailey, and Noble, 2011). ChoRE sequences regulating the *nsdhl* gene might be located at a more distal locus, or the regulation of *nsdhl* expression might be indirect and involve other transcriptional regulators downstream of MondoA.

**Author response image 1. sa2fig1:** Normalized luciferase expression in PAC2 zebrafish cells (n=4/condition) transiently overexpressing a 1kbp promoter region of the nsdhl gene in the absence (empty) or presence of mondoa and mlx overexpression in cells treated with 12 mM glucose.

Regarding the second question, we examined the induction of *txnipa* expression in *mondoa* morphants upon 2-DG injection. *txnipa* expression was not induced by 2-DG injection in the morphants. In addition, basal levels of *txnipa* expression were reduced in the morphants (see Figure 1—figure supplement 1G).

Reviewer #3:The study of Weger and coworkers explores the role of conserved intracellular sugar sensor MondoA-Mlx in zebrafish early development. The study shows that transient knockdown of MondoA and its necessary binding partner Mlx lead to impaired morphogenetic movement of epiboly in the embryos. The authors also use (likely null) mutants of MondoA and observe similar defects, however with very low penetrance. Rnaseq analysis of MondoA knockdown embryos revealed strong downregulation of nsdhl, encoding an enzyme involved in cholesterol biosynthesis. Knockdown of nsdhl showed similar epiboly phenotype as MondoA knockdown, implying that defect of cholesterol biosynthesis is the underlying reason for the developmental defect. This is supported by the finding that treatment with steroid hormone pregnenolone partially rescues the MondoA phenotype.This study is of high interest to the field, and therefore possibly suitable for eLife. It is to my knowledge the first characterization of MondoA-Mlx in vivo function in fish, it provides unprecedented evidence for MondoA-Mlx in early embryogenesis, and it establishes MondoA-Mlx as a putative regulator of steroid hormone metabolism. In general, the experiments have been carefully designed, conducted and documented and the conclusions are sound.I have a couple of requests to strengthen the central findings of the manuscript that should be addressed prior to publication.1) While nsdhl knockdown produces similar phenotype to MondoA knockdown, it remains unclear whether the epiboly phenotype can be mainly explained by deregulation of this one target gene. To test this, the authors should test whether transgenic nsdhl is sufficient to rescue the MondoA knockdown phenotype.

While *nsdhl* is the most downregulated gene identified in our RNA-seq study of the morphants, it is not the only one. In fact, the terpenoid backbone synthesis and the sterol biosynthesis pathways hold a prominent place among the pathways significantly downregulated in morphants (new Supplementary file 2). These data were illustrated in the gene set enrichment bar code plots of Figure 7 in the previous version of the manuscript, but not mentioned when we introduced the RNA-seq study of the morphants earlier in the manuscript. This may have created the impression that *nsdhl* is the only relevant target gene, which was not our intention. Our decision to focus on *ndshl* was motivated by the fact that it was the most downregulated gene in the entire set, and that it is a human disease gene. In the revised version of the manuscript, we have now expanded the description of the morphant RNA-seq data and mention the enrichment of the terpenoid and sterol biogenesis pathways within a systematic gene set enrichment analysis (Supplementary file 2 and Figure 7—figure supplement 1; we also indicate these pathways and the steroid biosynthesis pathway already in the simplified scheme in Figure 5B and mention them in the Abstract). Furthermore, we have rephrased parts of both the Results and the Discussion sections to better reflect these observations.

A rescue of *mondoa* morphants with *nsdhl* mRNA might seem to add a nice additional piece of data, but the rescue by the downstream pathway product pregnenolone is more informative in this context, especially as we observe deregulation also in other parts of the cholesterol/steroid biosynthesis pathway and not only in *nsdhl* expression. As described above and in our reply to the “Essential Revisions” section, Nsdhl is a crucial player in the pathway, but not the only one.

2) The partial rescue of MondoA knockdown by pregnenolone supports the idea that the phenotype is due to defects in steroid biosynthesis. The authors should directly analyze the levels of cholesterol (for example cholesterol binding fluorescent dyes) to test whether the biosynthesis is indeed affected by MondoA knockdown.

In an attempt to answer this question, we have tried to measure cholesterol related compounds both downstream and upstream of the Nsdhl-mediated synthesis steps applying mass spectrometry, but were not able to reliably detect them. One reason for this was interference by the large amount of cholesterol present at this early developmental stage in the embryonic extracts (personal communication from our collaborators, consistent with literature reports (Wiegand, 1996)). These high levels in early stage embryos are likely due to maternal deposition of cholesterol that potentially masks the effects of temporally or locally increased or decreased synthesis when measuring global concentrations. One can expect different pools of newly synthesized cholesterol and of cholesterol deposited in the egg. With regard to epiboly, our observations suggest that cholesterol synthesis seems to be required specifically in the YSL, since both MondoA and Nsdhl knockdowns restricted to the YSL impair epiboly (Figure 4C, D and 5G). Apparently, it is not the overall amount of cholesterol present in the yolk which is limiting. Rather, local synthesis or availability is important. Thus, local flux changes in cholesterol biosynthesis may be altered, which are currently not detectable due to a lack of adequate molecular probes.

3) RNAseq data should be more thoroughly analyzed and represented. Gene set enrichment analyses should be done to show, which pathways and processes are affected by MondoA knockdown. It would be also interesting to know, whether the conserved MondoA targets (shown in Figure 1) were affected by MondoA knockdown or MondoA mutant.

A global gene set enrichment analysis that complements the gene set enrichment analysis for the cholesterol related pathways shown in Figure 7 of the previous version of the manuscript has now been added to the manuscript (Supplementary file 2).

Expression of the conserved MondoA target genes was not or only slightly altered compared to wild-type in the morphant dataset (see Author response image 2); however, we observed that the glucose induction of *txnipa* was abolished by MondoA knockdown, as mentioned above in our reply to reviewer 2. In the mutant data set, *hk2* and *fasn* expression is not or only slightly altered compared to wildtype in both epiboly-affected and unaffected mutants. By contrast, expression of both *txnipa* and *b* is very low in both mutant phenotypes. The different behaviour of the *txnip* genes and of *hk2* and *fasn* may relate to redundancy with other factors capable of sustaining the expression of the latter two genes. Interestingly, similar observations were made in the skeletal muscle of mouse mutants of *mondoa*, with *txnip* being downregulated and *HKII* expression unchanged (Imamura et al., 2014).

**Author response image 2. sa2fig2:** Gene expression of MondoA targets in mondoa morphants and MZmondoa mutants.

4) The differences between MondoA knockdown and MondoA mutant are the most puzzling part of this manuscript. I have several requests regarding this part:– The authors performed an RNAseq of mutant embryos without an epiboly phenotype and state that "Surprisingly, nsdhl mRNA levels were not altered" (data not shown). It remains unclear to me whether this means not altered from MondoA knockdown (downregulated) or not altered from control (not downregulated). Please clarify this (and show the data). If not downregulated compared to control, it would be important to see what is the expression of nsdhl mRNA in mutants that show epiboly phenotype.

Thanks for pointing this out. We have now added the reference to the corresponding Figure (new Figure 7—figure supplement 2 E) and, in addition, provide the single plots for *nsdhl* in *mondoa* morphant and mutant embryos in the revised Figure 7—figure supplement 2F.

– The authors also show that a set of genes involved in terpenoid biosynthesis is differentially expressed in mutants (upregulated) and knockdown (downregulated) and suggest this would be the reason for the lacking phenotype. How are these genes expressed in MondoA mutant embryos with an epiboly phenotype? Does their knockdown cause an epiboly phenotype?

To answer this question, we now provide and analyze additional RNA-seq data of MZ*mondoa* mutants with affected epiboly. Please see our response to this point in the “Essential Revisions” paragraph above.

– In order to validate the effects of the genetic perturbations, a systematic comparison between the gene expression changes in the MondoA knockdown vs. mutants should be represented. Are other gene groups similarly deregulated in the two conditions and only the terpenoid group shows differential expression?

Common and differential gene expression changes in morphants and mutants are evident from our RRHO analysis, as detailed in the “Essential Revisions” section above. Furthermore, we have added a systematic metabolic pathway and GO term enrichment analysis comparing the different conditions to the manuscript (Supplementary file 2). This analysis shows, for example, that TCA pathway genes are upregulated in morphants and both mutant phenotypes, and that genes belonging to the lipopolysaccharide biosynthetic process GO are downregulated in all conditions. Not only the terpenoid group shows differential expression, but also, for example, sterol biosynthesis (see also Figure 7F).

Reviewer #4:[…] Major concerns:1) Rescue efficiencies by MondoA mRNA injected to MondoA morphants seem low.2) They tried to interprete the difference between MondoA morphants and knockout mutants by an unknown compensatory mechanism. I think the argument was not strong enough. The authors at least need to show that upregulation of the terpenoid pathway can rescue the epiboly phenotype seen in morphants.

The rescue efficiencies might appear low at a first glance, but are similar to those observed in previous studies that address other genes implicated in epiboly (Hsu et al., 2006; Schwend et al., 2011). Rescue efficiency may be limited by mRNA stability and/or translation efficiency of the rescue construct in vivo. Importantly, we observe a rescue not only with *mondoa* mRNA but also with the downstream product of the pathway, pregnenolone. Finally, the resistance of the mutant to the morpholino further validates the specificity of this tool for interference with MondoA function. Taken together, as also noted by reviewer #1, our rescue experiments satisfy “the recent criteria for MO experiments” (see also (Stainier et al., 2017)).

[Editors' note: further revisions were suggested prior to acceptance, as described below.]

The manuscript has been significantly revised. In the revised version, the authors extensively performed transcriptome analysis of morphants and mutants with or without the phenotype, and identified overlap between morphants and mutants and differences between mutants with and without the phenotype. Unexpectedly expression of nsdhl was not downregulated in the mutants with or without the phenotype. The authors found that mutants with unaffected epiboly showed stronger upregulation of genes in the terpenoid biosynthesis pathway and steroid hormone biosynthesis, compared to mutants with affected epiboly. The data strongly suggest that the upregulation of these genes compensated for Mondoa deficiency in the mutants, supporting the conclusion that MondoA-mediated cholesterol biosynthesis plays an important role.

We would like to thank the reviewers and editors for the overall positive evaluation of our work, in particular their appreciation that the data presented in the manuscript support “the conclusion that MondoA-mediated cholesterol biosynthesis plays an important role” in embryonic development.

Major concerns:1) A main concern with the original manuscript version was that there was no downstream mechanism for the epiboly phenotype presented, which would be consistently supported by evidence obtained from both morpholino and mutant experiments. With the new RNAseq dataset the gene expression output of the morphants and the mutants with affected epiboly can be compared. The morphants show significant downregulation of genes in the terpenoid backbone and sterol biosynthesis pathways (including the most strongly downregulated gene nsdhl), supporting the main conclusion. The mutants with affected epiboly, however, show no change in sterol biosynthesis (including nsdhl) or the terpenoid backbone biosynthesis pathways. The authors draw attention to the upregulation of the terpenoid backbone biosynthesis pathway in the unaffected mutants (in contrast to the affected mutants) and imply that these genes have a compensatory role. However, experimental evidence for such a compensatory effect is not presented. It also remains unclear to me what would these genes compensate for, since no impairment of cholesterol or steroid hormone metabolism is observed in the mutants. Accordingly, this should be discussed.

We have now expanded our discussion of the compensatory mechanisms in the Discussion section accordingly:

“Interestingly, many terpenoid, sterol and steroid biogenesis genes showed lower than wild-type levels of expression in the morphants and equal or higher levels in the mutants with unaffected epiboly, suggesting that increased steroidogenesis activity rescues epiboly in a majority of MZ mutant embryos. In the remaining mutant embryos, the lack of coordinated upregulation of terpenoid backbone biosynthesis genes is indicative of an incomplete compensation attempt. Apparently, concerted changes across the entire cholesterol/steroid biosynthesis pathway are needed to rescue epiboly progression in the absence of MondoA function. The mere restoration of *nsdhl* expression levels, for example, appears not to be sufficient for such a rescue, perhaps because a few genes in the sterol biosynthesis pathway still show deregulation of expression in MZ mutant embryos of both phenotypes (Figure 7—figure supplement 1 A). Therefore, even though globally there is no coherent up- or downregulation in sterol biosynthesis gene expression of mutants compared to wild-type embryos based on gene set enrichment analysis, pathway function in mutants may still be suboptimal and require the compensatory upregulation in the upstream terpenoid pathway to achieve sufficient synthesis flux. Furthermore, random variation in the amount of material deposited by the mother or in protein expression of key genes may limit compensation in the mutants with affected epiboly. A more detailed understanding of these processes awaits the determination of metabolite fluxes in morphants, mutants and wild-type embryos, specifically in the YSL.”

Further, the main conclusion is "regulation of cholesterol biosynthesis by glucose-sensing MondoA is required for zebrafish epiboly” is not sufficiently supported by the mondoa mutant data presented. From this aspect, the title better be modified.

In order to more precisely describe our observations, we have now modified the title to “MondoA regulates gene expression in cholesterol biosynthesis-associated pathways required for zebrafish epiboly”.

2) (similar to the #1) The fact that expression of nsdhl was not altered in the MZ mutants with or without the phenotype should be discussed (in Discussion).

As included above (reply to #1), this is now discussed explicitly:

“The mere restoration of *nsdhl* expression levels, for example, appears not to be sufficient for such a rescue, possibly because a few genes in the sterol biosynthesis pathway still show deregulation of expression in MZ mutant embryos of both phenotypes (Figure 7—figure supplement 1 A).”

The authors should address why they do not think the transcriptome analysis of morphants were artifacts.

We are convinced that the morpholino results are not artifacts. The most important control experiment we performed is the experiment that reveals resistance of the MZ*mondoa* mutants to morpholino-induced phenotypes (Figure 7D). If the observed epiboly were due to off-target effects of the morpholino, these off-target effects would still elicit an aberrant epiboly phenotype in MondoA mutants. To better highlight this point, we have now emphasized this aspect in the revised version of the manuscript as follows: “Indeed, we observed a resistance of the mutants to the perturbance of epiboly caused by this MO in wild-types (Figure 7D). This observation strongly suggests the presence of strong compensatory or buffering mechanisms that enable embryonic development even when MondoA function is genetically perturbed. It also further confirms the specificity of the morphant phenotype.”

In addition to this experiment, which is considered as “the most definitive evidence for MO specificity” (Stainier et al., 2017), we also provided additional controls to prove morpholino specificity in the manuscript. Specifically, we 1) employed a 5 bp mismatch MO control, which is a more stringent control than the frequently used GeneTools standard control morpholino; 2) we tested for knockdown efficiency with a reporter construct, 3) we could partially rescue the phenotype with a non-targetable mRNA., 4) we demonstrated that no p53 mediated off target effects occur by co-injection with an anti-p53 morpholino, 5) knockdown of genes coding for an interaction partner of the pathway (*mlx*) or for a downstream target gene (*nsdhl*) led to similar phenotypes, strongly suggesting their contribution to a common pathway, and 6) the morphant phenotypes of both *mondoa* and *nsdhl* could be partially rescued by treatment with the downstream effector of the pathway, pregnenolone.

Finally, regarding the transcriptome analysis, we observed that none of the eight genes recently reported as commonly upregulated off-target genes in morpholino experiments (*isg20*, *isg15*, *tp53*, *casp8*, *phlda3*, *gadd45aa*, *mdm2*; (Lai, Gagalova, Kuenne, El-Brolosy, and Stainier, 2019)) were upregulated in our morphant RNAseq data set: 6 were not significantly regulated, while two (*isg20*, *phlda3*) showed downregulation in the morphants. Furthermore, GO terms related to stress response and apoptosis were not enriched in our gene set enrichment analysis of the morphant data (Supplementary file 2), in contrast to observations in the cited study. We are therefore confident that our morphant RNAseq data reflect the specific knockdown of MondoA function.

3) The section "MZmondoa mutants show compensatory changes in expression of cholesterol/steroid biosynthesis genes" was really difficult to read. I think the authors better describe the results as they described in the reply mail to us.

We have now replaced the corresponding paragraphs in the Results section with the more comprehensive description from the previous letter, and we have expanded the morphant results and included it in Figure 7—figure supplement 2 as panel G.

Also subsection “Maternal-zygotic mutants of mondoa show partially penetrant epiboly phenotypes”: “an epiboly phenotype” should be “the epiboly phenotype”.

We have corrected this phrase as requested.

Subsection “MZmondoa mutants show compensatory changes in expression of cholesterol/steroid biosynthesis genes”: “MZmondoa embryos with and without an epiboly phenotype”….This should be: MZmondoa embryos with and without the mutant phenotype, or MZmondoa embryos with severe and no mutant phenotype or MZmondoa embryos with severe and no epiboly anomalies

We have changed this phrase to “To begin to explore potential compensatory mechanisms allowing epiboly progression in MZ*mondoa* mutants, we performed total RNA sequencing of MZ*mondoa* embryos with severe and no epiboly anomalies and of wild-type embryos to determine differential gene expression (Supplementary file 1).” We also altered “with and without epiboly phenotype” in the following sentences to “with and without aberrant epiboly phenotype” and made analogous replacements throughout the manuscript.

Also in the same section: “both phenotypes of MZ mutants” should be “the MZ mutants of both phenotypes”

This phrase has now been replaced by the more comprehensive description based on our previous letter, as requested under #3 above.

“…as an important component of the compensation response”. This is a compensation response specifically required to restore the phenotype.

The sentence now reads: “This key difference highlights the upregulation of gene expression in the terpenoid backbone biosynthesis pathway as a compensation response specifically required to restore the normal epiboly phenotype.”